# Observation of phonon Stark effect

Zhiheng Huang[1,2,10], Yunfei Bai[1,2,10], Yanchong Zhao[1,2], Le Liu[1,2], Xuan Zhao [1,2], Jiangbin Wu [3], Kenji Watanabe [4], Takashi Taniguchi [5], Wei Yang [1,2], Dongxia Shi[1,2], Yang Xu [1,2], Tiantian Zhang [6], Qingming Zhang[1,2,7], Ping-Heng Tan [3], Zhipei Sun [8], Sheng Meng [1,2,9], Yaxian Wang [1] ✉, Luojun Du [1,2] ✉ & Guangyu Zhang [1,2,9] ✉

Stark effect, the electric-field analogue of magnetic Zeeman effect, is one of the celebrated phenomena in modern physics and appealing for emergent applications in electronics, optoelectronics, as well as quantum technologies. While in condensed matter it has prospered only for excitons, whether other collective excitations can display Stark effect remains elusive. Here, we report the observation of phonon Stark effect in a two-dimensional quantum system of bilayer $2H$-MoS$_2$. The longitudinal acoustic phonon red-shifts linearly with applied electric fields and can be tuned over ~1 THz, evidencing giant Stark effect of phonons. Together with many-body ab initio calculations, we uncover that the observed phonon Stark effect originates fundamentally from the strong coupling between phonons and interlayer excitons (IXs). In addition, IX-mediated electro-phonon intensity modulation up to ~1200% is discovered for infrared-active phonon $A_{2u}$. Our results unveil the exotic phonon Stark effect and effective phonon engineering by IX-mediated mechanism, promising for a plethora of exciting many-body physics and potential technological innovations.

Stark effect, one of the renowned phenomena in modern physics, describes the energy shifting or splitting of spectra lines induced by external electric fields. It was first discovered in hydrogen atoms by Johannes Stark in 1913 and soon awarded with the Nobel Prize in Physics in 1919 for its remarkable contributions to quantum theory[1,2]. In condensed-matter physics, Stark effect has been demonstrated for excitons (i.e., bound pairs of electrons and holes) in various solid-state quantum systems, such as quantum dots, quantum wells and van der Waals heterostructures[2–15]. The emerging exciton Stark effect not only opens up innovative paradigms to control the material's properties and quantum states in a precise,

high-speed, reversible and efficient manner, but also creates unprecedented possibilities to underpin new physics and to introduce a rich variety of technological applications, such as on-chip electro-optical modulators[4,16], tunable quantum light sources[17,18], nanoscale spin rectifier control[19], and compact spectrometers[20]. Although notable progress has been witnessed in exciton Stark effect, the Stark effects of other solid-state collective excitations, such as phonons (i.e., the quantized vibrational excitations of a crystal lattice) that are essential for plenty of emergent physics and innovative applications (e.g., superconductivity, ultrafast carrier dynamics, nonequilibrium phenomena, ultrafast control of

[1]Beijing National Laboratory for Condensed Matter Physics; Key Laboratory for Nanoscale Physics and Devices, Institute of Physics, Chinese Academy of Sciences, Beijing 100190, China. [2]School of Physical Sciences, University of Chinese Academy of Sciences, Beijing 100190, China. [3]State Key Laboratory of Superlattices and Microstructures, Institute of Semiconductors, Chinese Academy of Sciences, Beijing 100083, China. [4]Research Center for Functional Materials, National Institute for Materials Science, 1-1 Namiki, Tsukuba 305-0044, Japan. [5]International Center for Materials Nanoarchitectonics, National Institute for Materials Science, 1-1 Namiki, Tsukuba 305-0044, Japan. [6]CAS Key Laboratory of Theoretical Physics, Institute of Theoretical Physics, Chinese Academy of Sciences, Beijing 100190, China. [7]School of Physical Science and Technology, Lanzhou University, Lanzhou 730000, China. [8]QTF Centre of Excellence, Department of Electronics and Nanoengineering, Aalto University, Tietotie 3, FI-02150 Espoo, Finland. [9]Songshan Lake Materials Laboratory, Dongguan, Guangdong Province 523808, China. [10]These authors contributed equally: Zhiheng Huang, Yunfei Bai. ✉e-mail: yaxianw@iphy.ac.cn; luojun.du@iphy.ac.cn; gyzhang@iphy.ac.cn

magnetism, and thermal transistors)[21–25], though highly desired, have thus far remained elusive.

Herein, we report the first observation of the Stark effect for phonons in a two-dimensional (2D) quantum solid of bilayer 2H-MoS₂. Specifically, the longitudinal acoustic (LA) phonon mode in bilayer 2H-MoS₂ undergoes a linear redshift with external electric fields when the interlayer exciton (IX) energy is tuned across its emission line, evidencing the first-order (also dubbed as linear) Stark effect of phonons. Remarkably, the observed phonon Stark effect in bilayer 2H-MoS₂ is giant and can reach an extremely large frequency change up to ~33 cm⁻¹ (~1 THz). We remark that although the control of phonons with gating has been reported in many 2D systems, such as monolayer/bilayer/trilayer graphene[26–30], monolayer transition metal dichalcogenides (TMDs)[31,32], and black phosphorus[33], the electrostatic doping effects, rather than electric field, are typically focused, and the phonon energy modulation is generally nonlinear and within 10 cm⁻¹. Guided by many-body first principles calculations, we pinpoint the underlying microscopic origin of the observed giant phonon Stark effect to be the strong coupling between phonons and highly tunable IXs. Furthermore, we demonstrate the electro-phonon modulation of emission

intensity mediated by IXs reaching as large as ~1200% for the infrared-active phonon mode $A_{2u}$. Our results demonstrate the emerging giant phonon Stark effect and effective electric control of phonon states by IX-mediated mechanism, holding a great promise for a rich diversity of emergent quantum phenomena and potential technological applications, such as electric-field-tunable phonon laser, dynamical control of heat transport and THz acoustic-electronic/optic devices.

## Results

### Linear Stark effect of IXs in bilayer 2H-MoS₂

High-quality, hexagonal boron nitride (h-BN) encapsulated dual-gate 2H-phase bilayer MoS₂ devices are fabricated by a van der Waals mediated dry layer-by-layer transfer-and-stack technique[34,35]. Figure 1a illustrates schematically the typical device structure (also see Supplementary Fig. 1 for the optical microscope images). Few-layer graphene is used as both the bottom and top gate electrodes to independently tune the out-of-plane electric field $F_z$ and carrier density $n_0$. Here $F_z = (C_b V_{bg} - C_t V_{tg})/2\varepsilon_0\varepsilon_{BL}$ and $n_0 = (C_b V_{bg} + C_t V_{tg})/e$, where $e$ is the elementary charge, $\varepsilon_0$ denotes the vacuum permittivity, and $\varepsilon_{BL}$ is the out-of-plane dielectric constant of bilayer MoS₂. $C_b(V_{bg})$ and $C_t$

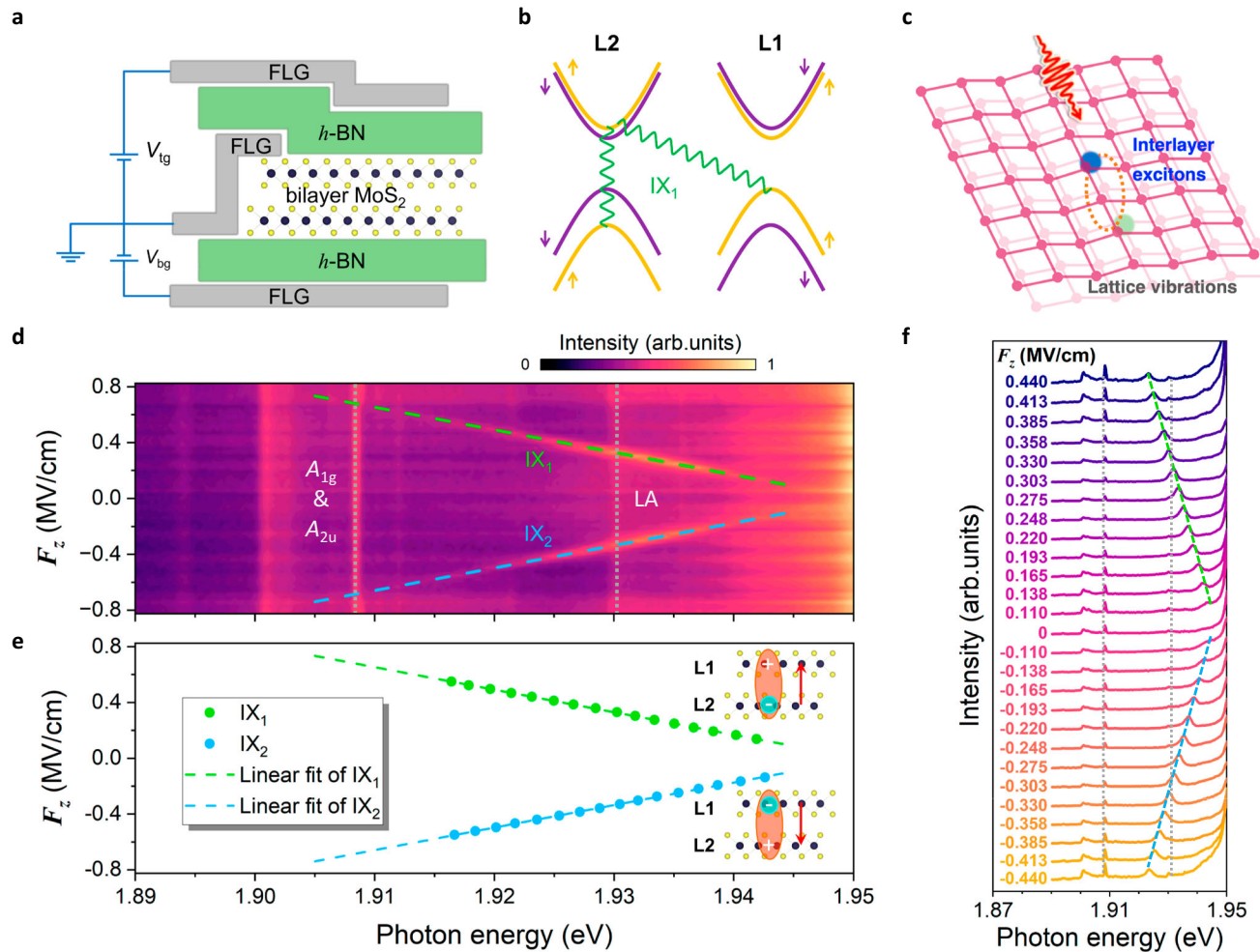

**Fig. 1 | Quantum-confined Stark effect of IXs in bilayer 2H-MoS₂. a** Schematic image of h-BN encapsulated dual-gate bilayer MoS₂ devices. Few-layer graphene (FLG) is used as both the bottom and top gate electrodes to tune the $F_z$. **b** Schematic of IX configurations with a strong intralayer B exciton component from lower layer L2. **c**, Schematic of coexistence of IXs and phonons. **d** Contour plot of the PL spectra of a typical device H1 as a function of photon energy (bottom axis) and $F_z$ (left axis). Doping density remains unchanged. IXs (Raman peaks) are highlighted by the dashed (dotted) lines. **e** Extracted emission energies of IX₁

(green disks) and IX₂ (blue disks) from (**d**) as a function of $F_z$. The green and blue dashed lines are linear-fits of IX₁ and IX₂, respectively. Insets: schematic of the IX configurations in real-space. The directions of the dipole moment denoted by red arrows depend on the location of the constituent electron, either in the bottom or top layer. **f** Normalized linecuts of PL spectra at selected $F_z$ from (**d**). Offset is set vertically for clarity and the corresponding $F_z$ of each curve is labelled. Green and blue dashed lines respectively represent IX₁ and IX₂ to guide for the eye. Grey dotted lines denote Raman modes.

($V_{tg}$) are the geometrical capacitances per area (applied voltages) for the bottom and top gates, respectively (see Methods for more details).

In contrast to the monolayer case, bilayer 2*H*-MoS$_2$ shows layer degree of freedom and can host IXs whose constituent electrons and holes are spatially displaced and thus are highly electric-field tunable by the first-order Stark effect[15,35–38]. It is noteworthy that because of the layer-hybridized hole states, IXs in 2*H*-stacked MoS$_2$ bilayers interact strongly with intralayer B excitons and acquire appreciable oscillator strengths (Fig. 1b)[15,35]. This is in stark contrast to IXs in typical TMD heterobilayers, where their coupling to light is substantially reduced[5,39]. Such a powerful combination of strong light-matter interaction and highly efficient electric tunability enables IXs to couple with other elementary excitations e.g., lattice vibrations (Fig. 1c), and potentially yields new hybrid excited states[18,39–41].

To capture the unique IXs in bilayer 2*H*-MoS$_2$, we perform the photoluminescence (PL) measurements as a function of the out-of-plane electric field $F_z$, while keeping the carrier density $n_0$ unchanged. Unless otherwise noted, all measurements (including both PL and Raman spectroscopy) are carried out in a high vacuum at 10 K with an on-resonance 633 nm laser excitation. Figure 1d depicts the colour plot of PL spectra against $F_z$ for a typical bilayer 2*H*-MoS$_2$ device H1 through a grating of 600 gr/mm. Clear features of IX emissions whose energies shift linearly with increasing $F_z$ can be unequivocally identified (dashed lines in Fig. 1d) and further confirmed by the linecuts at different $F_z$ (dashed lines in Fig. 1f). The linear IX energy shift with $F_z$ suggests the first-order Stark effect caused by the out-of-plane static electric dipole moments across the bilayer 2*H*-MoS$_2$[15,35]. Notably, there are two well-separated IX branches with opposite static electric dipole moments, i.e., IX$_1$ (IX$_2$) species with positive (negative) electric dipole moment. This can be well understood as the layer degeneracy of band structure in bilayer 2*H*-MoS$_2$, which gives rise to one IX with electron localized in the lower layer (i.e., IX$_1$) and the other with electron restrained in the upper layer (i.e., IX$_2$), as schematically depicted in the inset of Fig. 1e[15,35–37]. The IX$_1$ and IX$_2$ transition energies against applied electrical fields extracted by Voigt function fitting are respectively plotted as green and blue disks in Fig. 1e. The linear dependences (dashed lines in Fig. 1e) give the electric dipole moments of IX$_1$ and IX$_2$ to be $\mu(IX_1) = (0.612 \pm 0.003)e \cdot nm$ and $\mu(IX_2) = -(0.608 \pm 0.003)e \cdot nm$, respectively. This is in good agreement with the calculated value (e.g., $\pm 0.606\ e \cdot nm$) based on hybridized hole model (Supplementary Note 2) and previous results[15,35–37,42].

## Observation of phonon Stark effect

Apart from the electrically tunable IXs, several narrow peaks in the energy range from -1.939 to -1.90 eV, corresponding to the Raman shift from -166 to-480 cm$^{-1}$, are also noticeable in Fig. 1d, f (grey doted lines). In the light of prior work[43–46], these peaks can be recognized as the Raman phonon signals in bilayer 2*H*-MoS$_2$ (e.g., LA around 230 cm$^{-1}$, and $A_{2u}$&$A_{1g}$ around 406 cm$^{-1}$). Strikingly, the IXs can be tuned to cross these phonon lines by the electric fields. This may enable the resonant coupling between these phonons and IXs, giving rise to the formation of exciton-dressed phonon states with a non-vanishing electric dipole moment and thus exotic Stark effect of phonons. To better distinguish the fine features of these phonon lines and demystify the phonon Stark effect, we perform the $F_z$-dependent Raman measurements with an improved energy resolution through a grating of 1800 gr/mm. Figure 2a presents the colour plot of the first-order derivative of Raman intensity $I$ over phonon frequency $\omega$ ($\partial I/\partial\omega$) for device H1 (please refer to Supplementary Fig. 3 for the intensity plot of raw Raman spectra). Surprisingly, when IXs (navy dashed lines) are tuned by the electric field $F_z$ across the phonon line around 230 cm$^{-1}$, an exotic phonon mode (labelled as Stark phonon, SP) emerges and redshifts linearly with increasing $F_z$ (black dotted lines). In close resemblance to the linear Stark effect of atom spectra lines and excitons in solid-state quantum systems[1,2], such a linear modulation of

phonon energies by external electric fields offers the hallmark of first-order phonon Stark effect. We highlight that although vibrational Stark effect has been previously uncovered in molecule systems[47], this is the first achievement of linear Stark effect of phonon collective vibrations in condensed matter solids. Note that according to previous results[43] and our calculated results (Fig. 3) which we shall come to shortly, the intriguing SP mode with energy around 230 cm$^{-1}$ can be discerned as a LA phonon with finite momentum around M point.

To further verify the linear Stark effect of LA(M) phonon in bilayer 2*H*-MoS$_2$, we carry out $F_z$-dependent Raman spectroscopy measurements on other two *h*-BN-encapsulated bilayer 2*H*-MoS$_2$ devices (labelled as H2 and H3). Figure 2b, c respectively show the colour plot of the raw Raman data and $\partial I/\partial\omega$ for device H2 ranging from 166 cm$^{-1}$ to 270 cm$^{-1}$ (refer to Supplementary Note 4 for the results of device H3). Similar to what is observed in device H1 (Fig. 2a), an unconventional SP mode also emerges in devices H2 and H3 and displays linear energy modulation with $F_z$ (highlighted by the dotted lines in Fig. 2b, c and Supplementary Fig. 4), evidencing the first-order phonon Stark effect. Figure 2e presents the Raman linecuts at different $F_z$ where the linear redshift of the SP mode on $F_z$ is illustrated by the bold black dashed line. Fitting the Raman spectra of device H2 with the Voigt function gives the energies of SP mode against $F_z$, as shown in Fig. 2d (red squares). The perfect linear dependence of the energies of SP mode on the strength of electric field quantitatively confirms the first-order phonon Stark effect. We extract, via linear fitting (black dotted line in Fig. 2d), that the slope of the phonon Stark effect, which can be viewed as the 'Stark tuning rate' or 'phonon dipole', is $289 \pm 3$ cm$^{-1}$/(V/nm). Importantly, the observed 'phonon dipole' in bilayer 2*H*-MoS$_2$ is giant, more than an order of magnitude larger than the state-of-the-art results [-20 cm$^{-1}$/(V/nm)] for vibrational modes of molecule systems[47]. Benefiting from such a colossal 'phonon dipole', the electric modulation of phonon energies by the Stark effect can reach a frequency shift up to -33 cm$^{-1}$ (-1 THz) within the maximum electric fields permitted by our experiments (i.e., -1.587 MV/cm). We highlight that such a phonon Stark shift of -1 THz is fairly large, competitive to the best phonon energy modulation by other mechanisms (typically < 0.3 THz), such as phase control[48], symmetry engineering[49], Kohn anomaly mechanism[26,29], strain deformation[50], and optical/magnetic control[51,52]. Furthermore, a much larger phonon Stark shift can be anticipated by designing new device architectures, for example double ionic gated transistors which can enable the application of electric fields of more than one order of magnitude stronger (-30 MV/cm)[53].

## Underlying mechanism of phonon Stark effect

The observation of linear phonon Stark effect in bilayer 2*H*-MoS$_2$ is fairly surprising, considering that the direct coupling between phonons and electric fields is generally ignorable. We notice that the SP phonon mode begins to redshift linearly with the applied electric field $F_z$ when the IX emission line is tuned to resonate with it (Fig. 2a–c). This strongly indicates that the observed phonon Stark effect could be mediated by the electrically tunable IX states. To confirm the role of IXs in the phonon Stark effect, we perform $F_z$-dependent Raman spectroscopy measurements on a high-quality, encapsulated, dual-gate bilayer 3*R*-MoS$_2$ with the same device architecture shown in Fig. 1a (see Supplementary Fig. 10 for the optical image). Bilayer 3*R*-MoS$_2$ exhibits similar electronic structure and phonon dispersion to the bilayer 2*H* counterpart, but lacks the hybridized IXs as a result of layer-dependent Berry phase winding[54,55]. Remarkably, no apparent phonon Stark effect is observed in bilayer 3*R*-MoS$_2$ (please see Supplementary Note 10 for more details). Such a strong contrast between the bilayer 2*H*- and 3*R*-MoS$_2$ manifests the pivotal role played by IXs.

To explicitly unravel the microscopic origin of the observed linear phonon Stark effect, we carry out many-body first principles *GW* calculations (*GW*, one-body Green's function *G* and the dynamically screened Coulomb interaction *W*) in combination with the Bethe-

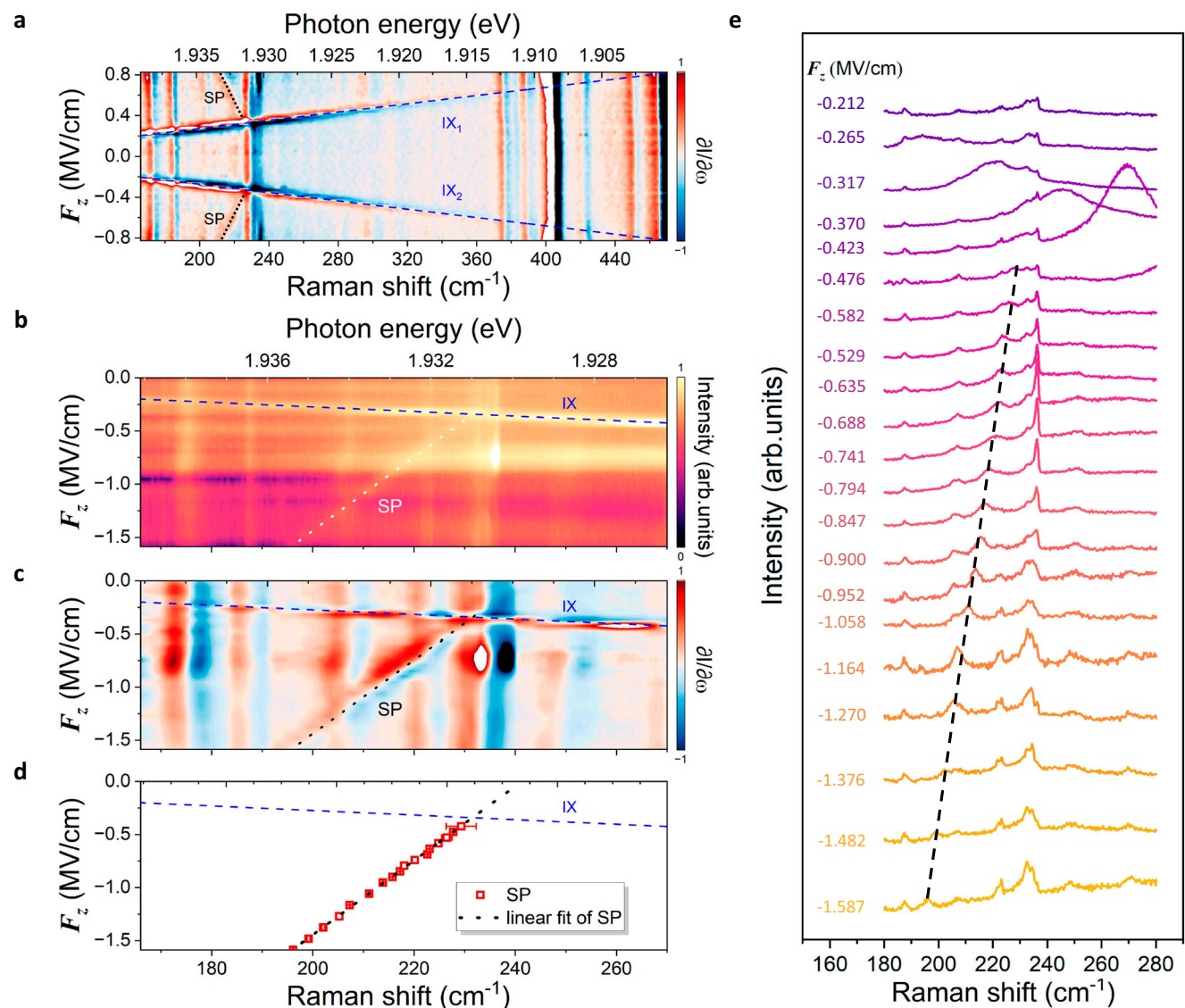

**Fig. 2 | Observation of phonon Stark effect. a** Colour plot of the first order derivative of phonon emission intensity $I$ over phonon energy $\omega$ ($\partial I/\partial \omega$) for device H1. Black dotted lines highlight the SP mode to guide for the eye. Navy dashed lines respectively represent the IXs. **b,** Contour plot of the Raman spectra of device H2 as a function of phonon energy (bottom axis) and electric field $F_z$ (left axis). **c** First-order derivative of (**b**). **d** Extracted phonon energy of SP mode (red squares) from (**b**) as a function of $F_z$. The error bars are dereved from the fitting. Dotted lines in **b**–**d** represent the linear-fitting of SP mode. Navy dashed lines in **b**–**d** represent the fitted IX in device H2. **e** Normalized linecuts of Raman spectra at selected electric field $F_z$ from (**b**). Offset is set vertically for clarity and the corresponding $F_z$ of each curve is labelled. Black dashed line traces SP mode to guide for the eye.

Salpeter equation (*BSE*) as well as density functional perturbation theory (DFPT). By expanding the total energy of the coupled exciton-phonon system in a perturbation theory-based formula and applying the variational principles, we can self-consistently solve the eigen-functions in the exciton and phonon basis, and obtain the mode- and momentum-resolved exciton-phonon coupling matrix element (see Methods and Supplementary Note 14 for more details). Figure 3a shows the calculated exciton absorption spectrum of bilayer 2*H*-MoS₂, where the well-defined intralayer exciton transitions (1s/2s state of intralayer A exciton: ~1.92/2.08 eV; 1s state of intralayer B exciton: ~2.12 eV) are nicely captured. Importantly, a prominent peak around 2.00 eV (labelled by the grey arrow), corresponding to the IX transition, can be unequivocally identified, in agreement with our measurements and previous studies[15,36,54]. The real-space wave functions of the constituent electrons (upper panel) and holes (lower panel) for a selected IX species are presented in Fig. 3b. Interestingly, the wave function of its constituent electron is completely localized in the upper layer, while the constituent hole mainly distributes in the lower layer.

Such spatially displaced wave functions of the constituent electrons and holes confirms the IX nature.

Figure 3c displays the calculated phonon dispersion of bilayer 2*H*-MoS₂, consistent with previous results[43]. In total, there are 18 phonon modes, which are labelled as modes 1–18 in order of increasing energy. Their coupling strength with the IXs, $\widetilde{G}_{ex-ph}$, are calculated and presented in Fig. 3d, for selected wavevectors, namely at the high symmetry points of the Brillouin zone (Γ, K and M). Strikingly, the giant coupling strength $\widetilde{G}_{ex-ph}$ is found between IXs and the mode 5/6 at zone boundary M point, i.e., the nearly degenerate LA(M) modes around 230 cm⁻¹ marked by the blue box in Fig. 3c (the corresponding normal displacements are shown in the upper panel of Fig. 3e). The lower panel of Fig. 3e displays the calculated $\widetilde{G}_{ex-ph}$ distribution of LA phonon in the momentum space, further confirming the strong coupling between LA(M) phonon mode and IXs. Such gigantic coupling could lead to intriguing collective excitations described by hybrid phonon-IX excited states, in which the phonon is 'dressed' with an exciton cloud[56]. This exciton-dressed phonon elementary excitation

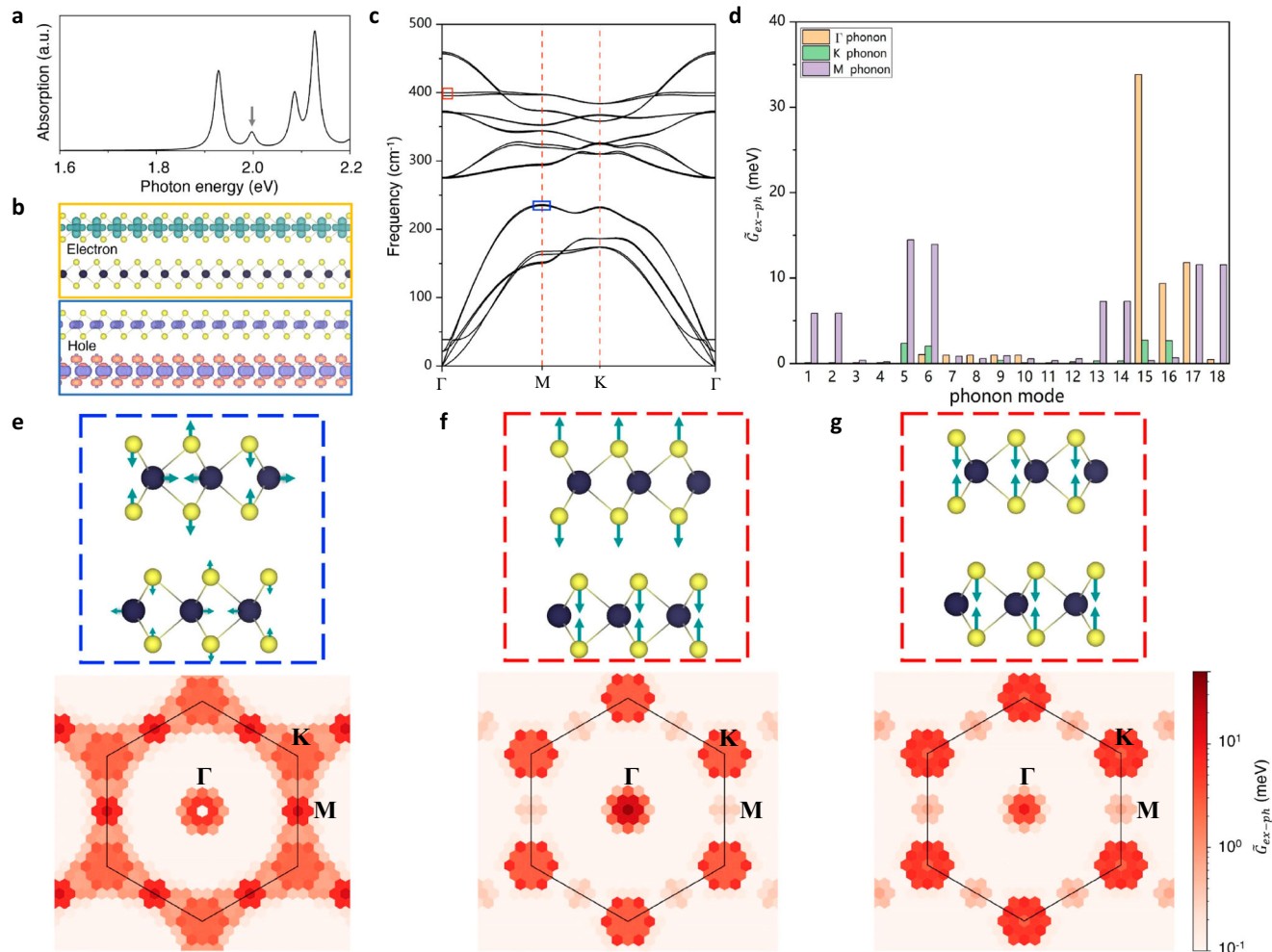

**Fig. 3 | Strong coupling between phonon and IXs. a** Calculated exciton absorption spectra of bilayer $2H$-MoS$_2$. Four main transitions are observed: 1 s/2 s state of intralayer A exciton at ~ 1.92/2.08 eV, 1 s state of intralayer B exciton at ~ 2.12 eV, and IX transition around 2.0 eV (labelled by grey arrow). **b** The real-space distribution of wave functions of the constituent electrons (upper panel) and holes (lower panel) for a selected IX species. **c** Calculated phonon dispersion of bilayer $2H$-MoS$_2$. The phonon modes at M ($\Gamma$) point with the largest coupling to the IXs are marked by blue (red) box, corresponding to LA(M) mode [$A_{1g}(\Gamma)$ and $A_{2u}(\Gamma)$ modes].
**d** Calculated coupling strength $\tilde{G}_{ex-ph}$ between IXs and all the 18 phonon modes at high symmetry points of $\Gamma$, M and K with order numbers in sequence of increasing energies. **e–g** Lower panel: calculated $\tilde{G}_{ex-ph}$ distributions in momentum space for selected modes 5 (**e**), 15 (**f**) and 16 (**g**). Upper panel: the normal displacements of LA(M) (**e**), $A_{2u}(\Gamma)$ (**f**) and $A_{1g}(\Gamma)$ phonon modes (**g**).

may inherit a fraction of the static dipole moment from the IXs and therefore be continuously tuned through out-of-plane electric fields, giving rise to the linear phonon Stark effect. This is in good agreement with our experimental results that a SP mode emerges and undergoes a linear shift with applied electric fields when IXs are electrically tuned across its emission line (Fig. 2).

We highlight that exciton-phonon coupling has been one of the research frontiers of condensed matter physics since its foundation in the 1950s[57,58]. In 2D systems, exciton-phonon coupling has been extensively studied and is believed to underlie many intriguing physics, including but not limited to phonon-assisted dark-exciton formation[59,60], phonon-mediated valley depolarization[61,62] and activation of optically silent phonon[63,64]. However, previous work mainly focusses on the interaction between phonons and intralayer excitons. By contrast, our work reveals the exciton-phonon coupling involving highly tunable IXs, and uncovers the exotic phonon Stark effect.

### Electric control of phonon intensity mediated by IXs

Besides the phonon mode 5/6 at M point, our theoretical calculations show that mode 15/16 at $\Gamma$ point (marked by the red box in Fig. 3c),

corresponding to the infrared active $A_{2u}(\Gamma)$/Raman active $A_{1g}(\Gamma)$ phonons with normal displacements shown in the upper panels of Fig. 3f, g, also display colossal coupling strength $\tilde{G}_{ex-ph}$ with IXs (Fig. 3d and lower panels of Fig. 3f, g). Such strong interactions between $A_{2u}(\Gamma)/A_{1g}(\Gamma)$ and highly tunable IXs may trigger the renormalization of Raman scattering cross-section and activity, leading to electric control of these two phonon states. Indeed, we observe an efficient electro-phonon modulation of emission intensity of $A_{2u}(\Gamma)$/$A_{1g}(\Gamma)$ mediated by IXs in bilayer $2H$-MoS$_2$. Figure 4a displays the colour plot of Raman spectra against the electric field $F_z$ in the range from 320 to 480 cm$^{-1}$ for the high-quality device H2. Around 406 cm$^{-1}$, there are two distinguishable phonon modes in bilayer $2H$-MoS$_2$ because of the Davydov splitting[43], namely one Raman active mode $A_{1g}(\Gamma)$ at ~406.8 cm$^{-1}$ and one infrared active $A_{2u}(\Gamma)$ mode at ~402.6 cm$^{-1}$. Since $A_{2u}(\Gamma)$ mode is infrared active, its emission intensity should be extremely weak compared to the Raman active $A_{1g}(\Gamma)$ mode. This is indeed the case under small electric fields where the IX emission line is lying below the $A_{2u}(\Gamma)$ phonon line. By contrast, when IXs (navy dashed lines, Fig. 4a) are tuned to resonate with $A_{2u}(\Gamma)$ phonon line by the electric field $F_z$, the emission intensity of $A_{2u}(\Gamma)$ phonon dramatically increases (see Supplementary Fig. 9 for the linecuts at different $F_z$).

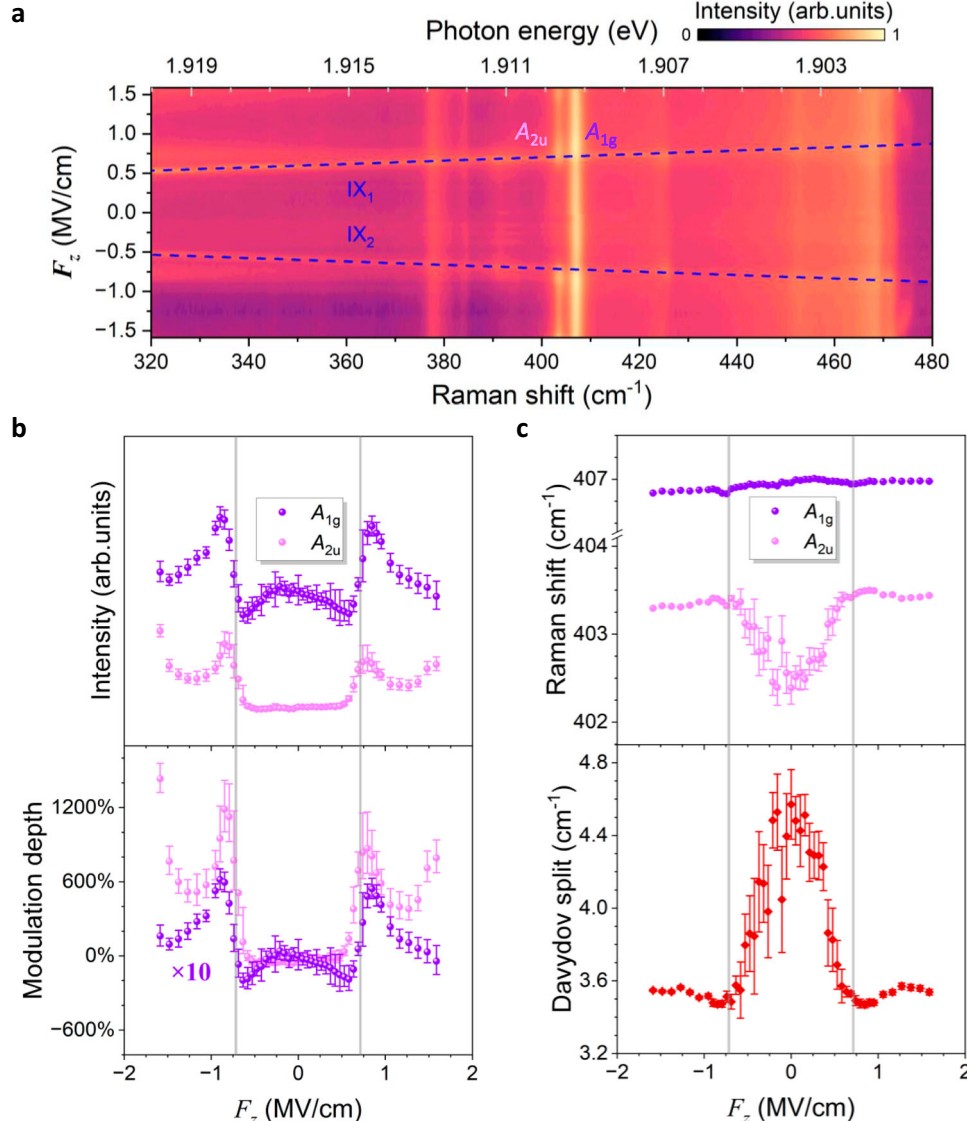

**Fig. 4 | Electric modulation of phonon intensity and Davydov splitting.**
**a** Contour plot of the Raman spectra of device H2 as a function of phonon energy (bottom axis) and $F_z$ (left axis). Navy dashed lines represent the IXs that are obtained by fitting. **b** Extracted phonon emission intensities (upper panel) and electric-phonon modulation depths (lower panel) as a function of $F_z$ for $A_{2u}(\Gamma)$ (pink spheres) and $A_{1g}(\Gamma)$ (violet spheres). Note that the modulation depths $\rho$ of

$A_{1g}(\Gamma)$ phonon are multiplied by a factor of 10 for better visualization. **c** Extracted phonon energies (upper panel) of $A_{2u}(\Gamma)$ (pink spheres)/$A_{1g}(\Gamma)$ (violet spheres), and their corresponding Davydov splitting (lower panel) against $F_z$. Grey vertical solid lines in (**b**) and (**c**) denote the specific electric field $F_0$ where the IX and $A_{2u}(\Gamma)$ emission lines coincidentally intersect. The error bars are derived from the fitting.

The upper panel of Fig. 4b shows the fitted emission intensities of infrared active $A_{2u}(\Gamma)$ phonon (pink spheres) and Raman active $A_{1g}(\Gamma)$ mode (violet spheres) as a function of $F_z$. Clearly, the emission intensities of infrared active mode $A_{2u}(\Gamma)$ experience a sharp enhancement around the specific electric fields $F_0 = \pm 0.715$ MV/cm (grey vertical lines) under which the IX emission line coincides with the $A_{2u}(\Gamma)$ phonon line. To quantitatively weigh their intensity change, we define the electro-phonon modulation depth as $\rho = \frac{I - I_0}{I_0}$, where $I$ ($I_0$) represents the phonon emission intensity at finite (zero) $F_z$. The lower panel of Fig. 4b presents the electro-phonon modulation depth $\rho$ of infrared active $A_{2u}(\Gamma)$ phonon (pink spheres) and Raman active $A_{1g}(\Gamma)$ mode (violet spheres) against $F_z$. The modulation depth $\rho$ of infrared active $A_{2u}(\Gamma)$ mode mediated by IXs is strikingly dependent on the applied electric fields and can be tuned in a wide range from ~0 to ~1200%. By contrast, there is no noticeable tuning of $A_{2u}(\Gamma)$ emission intensity in bilayer 3R-MoS$_2$ (Supplementary Note 10), again confirming the key role of IXs. In comparison, the maximum modulation depth of Raman

active $A_{1g}(\Gamma)$ mode is only ~60%, 20 times smaller than that of infrared active $A_{2u}(\Gamma)$ phonon. The much larger modulation depth $\rho$ of infrared active $A_{2u}(\Gamma)$ than Raman active $A_{1g}(\Gamma)$ can be understood as that owing to the antiphase displacements in adjacent layers (upper panel of Fig. 3f), infrared active $A_{2u}(\Gamma)$ mode can possess an interlayer electric dipole and thus a stronger coupling strength with the IXs (Fig. 3d)[64].

In addition to the intensity modulation, the energy of infrared active $A_{2u}(\Gamma)$ phonon and thus the Davydov splitting, i.e., the energy difference between $A_{2u}(\Gamma)$ and $A_{1g}(\Gamma)$ phonon modes, can also be engineered by the electric field, as depicted in Fig. 4c. While the $A_{1g}(\Gamma)$ phonon energy is insensitive to the electric field, the energy of $A_{2u}$ phonon blueshifts gradually as the electric field increases and reaches its maximum around $F_0$ (grey solid lines) at which the IX and $A_{2u}(\Gamma)$ emission lines resonate with each other (upper panel of Fig. 4c). The lower panel of Fig. 4c shows the Davydov splitting against the electric field, which follows the similar electric field-dependent evolution with the $A_{2u}(\Gamma)$ phonon energy and can be modulated by ~1.1 cm$^{-1}$.

## Discussion

In summary, we demonstrate the observation of linear phonon Stark effect in bilayer 2$H$-MoS$_2$ mediated by highly efficient gate-tunable IXs. The LA(M) phonon energy begins to redshift linearly with the applied electric fields when the IXs are tuned to resonate with its emission line, and the Stark shift can reach a frequency up to ~1 THz within the experimentally accessible electric field range. Together with many-body first principles calculations, we unveil that the strong coupling between the LA(M) phonons and IXs underlies the observed giant phonon Stark effect in bilayer 2$H$-MoS$_2$. In addition, IX-mediated strong renormalization of phonon emission intensity up to ~1200 % is achieved for infrared-active $A_{2u}$($\Gamma$) phonon mode. Our results demonstrate an IX-mediated mechanism for emerging phonon Stark effect and phonon engineering, and can also been applied to a wide range of solid-state quantum systems, such as TMD homo- and hetero-structures, promising the prospect of fascinating many-body physics and novel applications such as phonon lasers and THz acoustic-electronic devices.

## Methods

### Device Fabrication

$h$-BN encapsulated dual-gate devices were fabricated by a van der Waals mediated dry transfer technique. In short, bilayer MoS$_2$, few-layer graphene and $h$-BN, were first mechanically exfoliated from bulk crystals on 285 nm SiO$_2$/Si substrates. We highlight that all the bilayer 2$H$-MoS$_2$ (3$R$-MoS$_2$) samples we investigated are directly exfoliated from bulk 2$H$ (3 $R$) crystals (acquired from HQ Graphene), and the twist angle between the two constituent layers is a perfect 60° (0°). Flakes with appropriate size and thickness were then selected based on their optical contrast. Few-layer graphene is utilized as ground and gate electrodes. After stacking and releasing via layer-by-layer dry-transfer method, the devices were annealed in argon/hydrogen atmosphere at 350 °C for 4 h to diminish the influence of strains and bubbles, and improve quality. The thickness of gate dielectric material $h$-BN on each side is determined by atomic force microscope (AFM) before $F_z$-dependent optical measurements. Finally, metal contacts to ground and gate electrodes were patterned by the standard micro-fabrication processes, including e-beam lithography (EBL), metal evaporation Ti (3 nm)/Au (30 nm) and lifting-off.

### Determination of $F_z$ and $n_0$

The dual-gate device structure enables us to independently tune the vertical electric field ($F_z$) in bilayer MoS$_2$ without changing the doping density $n_0$. A parallel plate capacitor model is used to extract the $F_z$ under a top/bottom gate $V_{tg}/V_{bg}$. In this way, the displacement fields $D_T$ and $D_B$ across the top and bottom $h$-BN are: $D_T = C_T V_{tg}$ and $D_B = C_B V_{bg}$. $C_T = \frac{\varepsilon_0 \varepsilon_{h-BN}}{t_T}$ and $C_B = \frac{\varepsilon_0 \varepsilon_{h-BN}}{t_B}$ respectively represent the geometric capacitance per unit area of top and bottom gates. Here, $t_T$ ($t_B$) is the thicknesses of the top (bottom) $h$-BN layer as determined by AFM measurements, $\varepsilon_0$ is the vacuum dielectric constant and $\varepsilon_{h-BN} \approx 3.0$ is the out-of-plane dielectric constant of $h$-BN. The vertical electric field can be defined as $F_z = \frac{D}{\varepsilon_0 \cdot \varepsilon_{BL}}$, where $\varepsilon_{BL} \approx 6.5$ is the out-of-plane dielectric constant of bilayer MoS$_2$ and $D = \frac{1}{2}[D_T - D_B]$ is the electric displacement field in the system. Thus, we can calculate $F_z$ as $F_z = (C_B V_{bg} - C_T V_{tg})/2\varepsilon_0 \varepsilon_{BL} = \frac{\varepsilon_{h-BN}}{2 \cdot \varepsilon_{BL}} \frac{1}{t_T} \left[ V_{tg} - \frac{t_T}{t_B} V_{bg} \right]$. Here positive/negative $F_z$ represents vertical electric field upward/downward. Meanwhile, the doping density has a form as $n_0 = (C_T V_{tg} + C_B V_{bg})/e = \frac{\varepsilon_0 \varepsilon_{h-BN}}{e} \frac{1}{t_T} \left[ V_{tg} + \frac{t_T}{t_B} V_{bg} \right]$. Clearly, by keeping $V_{tg} + \frac{t_T}{t_B} V_{bg}$ as a constant and changing $V_{tg} - \frac{t_T}{t_B} V_{bg}$, we can tune $F_z$ while keeping $n_0$ unchanged.

### Optic measurements

In our cryogenic $F_z$-dependent optic experiments, devices were wire-bonded onto a chip carrier, placed in an optical chamber with a high vacuum and cooled down to 10 K by a closed cryocooler (CS-204PF-DMX-20B-OM from ARS). Raman (PL) spectra were obtained using a HORIBA spectrometer (LabRAM HR Evolution) in a confocal back-scattering configuration through a grating of 1800 (600) gr/mm. Light from 633 nm (1.96 eV) continuous laser with a power of 137 µW (34 µW) for Raman (PL) measurements was focused through a Nikon objective (N.A. = 0.5, W.D. = 10.6, F.N. = 26.5) onto the sample with a spot diameter of ~2 µm. The spectrometer integration time was 30 s (5 s) for Raman (PL) measurements.

### Many-body theoretical calculations

We perform ab initio calculations using Quantum Espresso code[65] combining density functional theory (DFT) and density functional perturbation theory (DFPT). Fully relativistic optimized norm-conserving Vanderbilt (ONCV) pseudopotentials[66] for the PBE exchange-correlation functional[67] were used for both Mo and S, which allowed us to calculate the electronic structure including spin-orbit coupling (SOC). To correctly account for interlayer coupling, dft-d2 correction are included. An $11 \times 11 \times 1$ Gamma centred $k$-mesh with 100 Ry cutoff is employed for the Brillouin zone integration and phonon dispersion is calculated on an $18 \times 18 \times 1$ $q$-mesh.

$G_0 W_0 + BSE$ calculation was carried out with Yambo code[68,69] with the same $18 \times 18 \times 1$ $k$-mesh in the Brillouin zone. A cutoff of 10 Ry of static screening and a total number of 400 bands are employed for the single-shot $G_0 W_0$ calculation to assure enough unoccupied states. A $G_0 W_0$ direct gap of 2.807 eV is obtained, in good agreement with previous results. Four valance bands and four conduction bands are included in $BSE$ calculation, from which we obtain the dispersion of eight exciton bands, and a binding energy of 0.879 eV for the A exciton. We further construct the mode and momentum-resolved exciton-phonon coupling matrix on the same grid.

## Data availability

The Source Data underlying the figures of this study have been deposited in Figshare and are available at https://doi.org/10.6084/m9.figshare.25751568. All raw data generated during the current study are available from the corresponding authors upon request.

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

## Acknowledgements

We thank Teng Yang for helpful discussions. This work is supported by the National Key R&D Program of China (2023YFA1407000, 2021YFA1202900, 2021YFA1400502), the National Science Foundation of China (NSFC) (12274447, 61888102), Guangdong Major Project of Basic and Applied Basic Research (2021B0301030002), and the Strategic Priority Research Program of Chinese Academy of Sciences under the grant No. XDB0470101. Y. W. and S. M. acknowledge funding support from the Ministry of Science and Technology (No. 2021YFA1400201), NSFC (Nos. 12025407, 92250303 and 11934004), and Chinese Academy of Sciences (Nos. YSBR047 and XDB33030100). K.W. and T.T. acknowledge support from the JSPS KAKENHI (20H00354, 21H05233 and 23H02052) and World Premier International Research Center Initiative (WPI), MEXT, Japan.

## Author contributions

L.D. and G.Z. supervised this work; L.D. and Z.H. conceived the project and designed the experiments; Z.H. fabricated the devices and carried out the optical measurements; Y.B. conducted the first principles calculations under the supervision of Y.W. and S.M.; K.W. and T.T. provided high quality *h*-BN crystals; X.Z. performed SHG measurements under the supervision of Y.X.; Z.H. and L.D. analysed the data; Y.Z., L.L., W.Y., D.S., J.W., T.Z., Q.Z., P.-H. T. and Z.S. helped with data analysis; Z.H., Y.W., L.D., and G.Z. co-wrote the manuscript. All authors discussed the results and commented on the paper.

## Competing interests

The authors declare no competing interests.
