## [Peer Review File · Nature Communications]

Observation of phonon Stark effectREVIEWER COMMENTS

Reviewer #1 (Remarks to the Author):

The manuscript combines theory and experiment to study a gated 2H MoS² heterostructure. They show strong evidence that there is a Stark effect on some phonon modes, which are coupled to the interlayer excitons. This is impressive work and a great read. It is surprising that physics would allow this, and it is a challenging project to show this. There is no doubt a great many follow up studies for this system and other 2D systems.

I recommend that this manuscript be published pending the few changes below. Although I give a positive review (great work), I respectfully ask the authors to work hard on the notes below because they are necessary (IMO) for a general audience to best understand the work and make it qualified for Nat Comm.

(1) What was the unit cell size of the MoS²? What twist angle did you assume? I am confident it was a 2H MoS² aligned, but a separate SI section that shows the heterostructure more specifically, along with more theory details, is needed for more audience clarity.

(2) Please add more background literature review about other manuscripts that gated a 2D heterostructure (or MoS²) and saw shifts in the Raman spectra. I don't think anyone has shown it in this detail or magnitude (a quick search did not reveal anything), but please concisely explain the current state of the field with respect to using gating to control the Raman modes.

(3) I did not fully understand how the coupling between the excitons and phonons can be determined using theory. Please have a summary explanation, and something longer in the SI. Please also go into more detail how the excitons were calculated. I am very impressed you could do those very complex calcs. I don't understand your method fully, and wish to evaluate it more precisely to understand which assumptions were made to make the calcs tractable. Truthfully, I am even more impressed that all this could be done with interlayer excitons, where the calcs must have been even more challenging.

(4) Dr. Prof. Emmanuel Rashba is a pioneer in the field of phonon-exciton coupling. Please add them to the citations. Along those lines, please have a concise paragraph that discusses exciton-phonon coupling in 2D systems. The fact that this is across the interlayer region is noteworthy. Exciton-phonon coupling in 2D systems is particularly interesting and your work builds onto this.

Reviewer #2 (Remarks to the Author):

The authors present a joint experiment-theory study on the appearance of Stark phonons in 2H-MoS² bilayers. These phonons appear as the IXs are tuned to resonate with the LA phonon emission line, and have a linear energy shift while tuning the applied out-of-plane electric field. Theoretical calculations are then carried out and find a strong coupling strength between IXs and LA phonons, which 'dresses' the phonons with exciton clouds and therefore fractions of electric dipole. Comparison with 3R- MoS² bilayer, where IXs and Stark phonons are missing, further proves their interpretation. Furthermore, a phonon intensity modulation as high as 1200% through an IX-mediated mechanism is observed as the IXs' energies are tuned to the A_{2u} phonon line.

The addressed topic is novel and is in the core of one of the most active research areas in condensed matter physics. The manuscript is well written and the main message is clearly conveyed. The experimental data is beautiful and the theoretical calculations contribute to a better understanding of the exciton-phonon coupling in TMDC materials. So I think the work has the potential to deserve publication in Nature Communications. There are, however, a few issues that the authors need to address to convince me to recommend the work for publication:

1) Why is MoS² A exciton (~1.92eV) missing from the spectra?

2) If possible, it would be great to test with different laser wavelengths/different TMDC materials (Mo-based is preferred) to check if the Stark phonons are gone when the LA-IX coupling is absent.

3) The theoretical calculation indicates a strong coupling between LA(M) phonon and IXs. Why is LA phonon's Raman intensity not modulated (to a high extent as A2u) through the same IX-mediated mechanism?

4) Similar question to 3), why do A1g and A2u not have phonon Stark effect as their couplings to IXs are also strong? If possible, it would be great to elaborate in the manuscript what might determine the results (SP or intensity modulation) of the phonon-IX coupling.

5) In figure 1d & 2a, the peak at $1.923\text{eV}/295\text{cm}^{-1}$ at higher electric field seems to share similar features with the intensity-modulated A2u phonon. Is this peak repeatable? (It seems absent from 3R- MoS₂. How about H2 and H3?)

Reviewer #3 (Remarks to the Author):

In this manuscript, Huang, et.al. explores an intriguing linear shift in energy and intensity modulation of phonon modes in bilayer 2H MoS₂ via an external electric field. The term **Phonon Stark Effect** is coined. Although experimentally detailed and computationally robust, the authors have failed to draw major insights based on their observation and certain technical aspects remain unclear. This makes it **lacking the content and the quality required for publication in Nature Communication in its current form**. Listed are few major technical queries regarding the results followed by some minor suggestions and general comments.

- What is the physical implication of a **DC Phonon Stark** effect in the bilayer MoS₂? In reference 38, vibrational Stark effect is introduced where the DC electric field modifies the intermolecular interactions via dipole-dipole interaction, which shows up as a vibrational energy renormalization. What is the counterpart in solid state system and why does it need an exciton to mediate it? All available theory (to my knowledge) on Stark effect for phonon dressed excitons refer to a AC Stark effect where exciton energy bands are renormalized due to an exchange type interaction [Example Phys. Rev. B 58, 1349, 1998, phys. stat. sol. (b), 150: 431-436, 1988]. But the effect described in the paper is renormalization of phonon energy state upon application of a DC field, which signifies some sort of material modification like stress or strain. Is that what is happening?

- What roles does the resonance condition play? When IX is tuned over the Raman line, is there really a resonance condition met? Energetically phonons are far away from excitons. The exciton is tuned between 1.91 and 1.94 eV, while the 230cm^{-1} LA phonon line has an energy of 29meV?

- As a follow up question to the previous question, in the experiments how is the spectral filtering of excitonic PL and Raman signal done, especially when they become resonant? Because of the GW calculation backing, it is very evident that there is large exciton phonon coupling for certain phonon modes. But the fact that the energy renormalization requires the exciton line to be resonantly tuned to a Raman line begs the following question: **Are we seeing some kind of optical Fano effect between the coherent Raman scattering signal and the PL from the exciton?** That would explain the red shift of the 230cm^{-1} Raman line compared to the blue shift of the IX1 and IX2 lines in cm^{-1} under applied DC bias F_z (Exciton red shifts and thus the energy delta between the Raman laser and the exciton peak increases and thus IX appear to blue shift in cm^{-1} scaling).

- An effective way to dispel my secondary explanation in the previous bullet point is to perform Raman with say 532nm excitation, where the 230cm^{-1} LA phonon would be far from the IX tuning range. Do we expect this phonon Stark affect to show up? If not, then we need to clarify the type of resonance condition we are describing, as the phonon states are always present in the solid at finite temperatures. And if we generate excitons, because of physical proximity and presence of exchange type interactions, the excitons might be dressed with phonons.

In other words, if spectral overlap of the IX PL and Raman is key to this phenomenon, then an explanation for how to exclude optical Fano effects need to be provided. Maybe this previous work will provide some insight. ACS Nano , 15, 9, 15371–15380, 2021

- Why do LA phonon and the A_{2u} phonon show different behavior upon DC field? LA phonon redshifts and the A_{2u} blueshifts.

Minor Suggestions

- Please include linewidths of the IX exciton PL in the text or supplementary information.
- In conclusion the work presented here currently **does not meet the standards required for publication in Nature Communication in its current form**. Some revisions along with more insightful discussions of this work is essential to be considered for publication.

RESPONSE TO REVIEWERS' COMMENTS

Reviewer #1 (Remarks to the Author)

The manuscript combines theory and experiment to study a gated $2H$ MoS₂ heterostructure. They show strong evidence that there is a Stark effect on some phonon modes, which are coupled to the interlayer excitons. This is impressive work and a great read. It is surprising that physics would allow this, and it is a challenging project to show this. There is no doubt a great many follow up studies for this system and other 2D systems.

I recommend that this manuscript be published pending the few changes below. Although I give a positive review (great work), I respectfully ask the authors to work hard on the notes below because they are necessary (IMO) for a general audience to best understand the work and make it qualified for Nat Comm.

Response 1:

We sincerely thank the Reviewer for the very positive evaluation on our manuscript and his/her recommendation for publication in *Nature Communications*. We also appreciate the Reviewer's insightful and constructive comments for improvement. Below we address the Reviewer's comments point by point.

(1) What was the unit cell size of the MoS₂? What twist angle did you assume? I am confident it was a $2H$ MoS₂ aligned, but a separate SI section that shows the heterostructure more specifically, along with more theory details, is needed for more audience clarity.

Response 2:

We thank the Reviewer for the kind comment. For all the bilayer $2H$ -MoS₂ samples we investigated, they are directly exfoliated from bulk $2H$ crystals (acquired from HQ Graphene), and the twist angle between the two constituent layers is a perfect 60° . The unit cell size is $\frac{\sqrt{3}}{2}a^2 = 8.65 \text{ \AA}^2$, where $a = 3.16 \text{ \AA}$ represents the lattice constant of bilayer $2H$ -MoS₂. In addition, we remark that for the bilayer $3R$ -MoS₂ samples we studied, they are also mechanically exfoliated from bulk $3R$ crystals and the twist angle between the two constituent layers is a perfect 0° .

For more clarity, we have added the following content in the revised manuscript (lines 281-284, page 7):

“We highlight that all the bilayer $2H$ -MoS₂ ($3R$ -MoS₂) samples we investigated are directly exfoliated from bulk $2H$ ($3R$) crystals (acquired from HQ Graphene), and the twist angle between the two constituent layers is a perfect 60° (0°)”.

(2) Please add more background literature review about other manuscripts that gated a 2D heterostructure (or MoS₂) and saw shifts in the Raman spectra. I don't think anyone has

shown it in this detail or magnitude (a quick search did not reveal anything), but please concisely explain the current state of the field with respect to using gating to control the Raman modes.

Response 3:

We thank the Reviewer for the kind suggestion. The modulation of phonon energy with gating has been reported in many 2D systems, such as monolayer/bilayer/trilayer graphene [*Nat. Mater.* **6**, 198 (2007); *Phys. Rev. Lett.* **101**, 136804 (2008); *Phys. Rev. Lett.* **101**, 257401 (2008); *Phys. Rev. B* 102, 165415 (2020); *Nat. Commun.* 15, 1888 (2024)], monolayer transition metal dichalcogenides [*Phys. Rev. B* **85**, 161403(R) (2012); *Phys. Rev. X* **9**, 031019 (2019)], and black phosphorus [*2D Mater.* **3**, 015008 (2016)]. However, these previous works typically employ single gate geometry and focus on the electrostatic doping effects. Furthermore, the phonon energy modulation with doping density is generally nonlinear and within 10 cm^{-1} . In our case, we focus on the effects of out-of-plane electric field by employing a dual-gated device architecture, and reveal a linear control of phonon energy up to $\sim 33 \text{ cm}^{-1}$.

Following the Reviewer's kind suggestion, we have added the following content to highlight the current state of the field with respect to using gating to control the Raman modes in the revised manuscript (lines 61-65, page 2):

“We remark that although the control of phonons with gating has been reported in many 2D systems, such as monolayer/bilayer/trilayer graphene²⁶⁻³⁰, monolayer transition metal dichalcogenides (TMDs)^{31,32}, and black phosphorus³³, the electrostatic doping effects, rather than electric field, are typically focused, and the phonon energy modulation is generally nonlinear and within 10 cm^{-1} ”.

(3) I did not fully understand how the coupling between the excitons and phonons can be determined using theory. Please have a summary explanation, and something longer in the SI. Please also go into more detail how the excitons were calculated. I am very impressed you could do those very complex calcs. I dont understand your method fully, and wish to evaluate it more precisely to understand which assumptions were made to make the calcs tractable. Truthfully, I am even more impressed that all this could be done with interlayer excitons, where the calcs must have been even more challenging.

Response 4:

We thank the Reviewer for recognizing our theoretical calculations and for the opportunity to provide a more detailed description of the theoretical approach used in our work.

Pertaining to the calculations of exciton absorption spectrum, all optical excitonic transitions are calculated using the state-of-art *GW-BSE* approach (*GW*, one-body Green's function *G* and the dynamically screened Coulomb interaction *W*; *BSE*, Bethe-Salpeter equation, which is available in multiple *ab initio* codes [*Comput. Phys. Commun.* 180, 1392 (2009); *Comput. Phys. Commun.* 183, 1269 (2012)] and can give the exciton wavefunction

at finite momentum [*J. Phys.: Condens. Matter* 31, 325902 (2019)]. Following the implementation in Yambo code, the Bethe-Salpeter equation can be reduced to an eigenvalue problem of the Hamiltonian H , i.e. the H_{BSE} in our implementation, as

$$H_{mm'k}^{nn'k} = (\varepsilon_{nk} - \varepsilon_{n'k})\delta_{nm}\delta_{n'm'}\delta_{kk'} + (f_{n'k} - f_{nk}) \begin{bmatrix} 2\bar{V}_{mm'k}^{nn'k} - W_{mm'k}^{nn'k} \end{bmatrix}. \quad (1)$$

Here both W and \bar{V} are integrals of the Bloch functions

$$W_{ss'k_1}^{nn'k} = \frac{1}{\Omega N_q} \sum_{\mathbf{G}\mathbf{G}'} \rho_{ns}(\mathbf{k}, \mathbf{q} = \mathbf{k} - \mathbf{k}_1, \mathbf{G}) \rho_{n's'}^*(\mathbf{k}_1, \mathbf{q} = \mathbf{k} - \mathbf{k}_1, \mathbf{G}') \times \varepsilon_{\mathbf{G}\mathbf{G}'}^{-1} v(\mathbf{q} + \mathbf{G}'),$$

$$\bar{V}_{ss'k_1}^{nn'k} = \frac{1}{\Omega N_q} \sum_{\mathbf{G} \neq 0} \rho_{nn'}(\mathbf{k}, \mathbf{q} = \mathbf{0}, \mathbf{G}) \rho_{ss'}^*(\mathbf{k}_1, \mathbf{q} = \mathbf{0}, \mathbf{G}) v(\mathbf{G}), \quad (2)$$

where N_q the number of points in the Brillouin zone (BZ) sampling, Ω the unit cell volume, and $v(\mathbf{q} + \mathbf{G}) = \frac{4\pi}{|\mathbf{q} + \mathbf{G}|^2} \cdot \varepsilon_{\mathbf{G}\mathbf{G}'}^{-1}$ is the random-phase approximation (RPA) dielectric function. One can see that the electron-electron scattering term (W) and the exchange interaction (\bar{V}) now has a combination of conduction and valence band index as ($|v\mathbf{k}\rangle$ the single particle level) $\rho_{nm}(\mathbf{k}, \mathbf{q}, \mathbf{G}) = \langle n\mathbf{k} | e^{i(\mathbf{q} + \mathbf{G})\mathbf{r}} | m\mathbf{k} - \mathbf{q} \rangle$. We include four valance bands and four conduction bands in BSE calculation, from which we obtain the dispersion of eight exciton bands, naturally embracing the interlayer exciton and the intralayer A/B exciton states of bilayer $2H$ -MoS₂.

To get the mode- and momentum-resolved coupling strength for the exciton-phonon coupling, we start from a perturbation theory-based framework, which has been applied to compute polaron formation [*Phys. Rev. Lett.* 122, 246403 (2019); *Phys. Rev. B* 99, 235139 (2019)], where the total energy of a system containing an extra electron can be described as a summation of its ground state energy and a linear expansion with respect to the lattice displacement

$$E\{\tau_{\kappa\alpha i}, \psi\} = E\{\tau_{\kappa\alpha i}^0, \psi_{n\mathbf{k}}^0\} + \frac{1}{2} C_{\kappa_1\alpha i, \kappa_2\beta j} \tau_{\kappa_1\alpha i} \tau_{\kappa_2\beta j} + \int \psi^* \left[H_{KS}^0 + \frac{\partial V_{KS}^0}{\partial \tau_{\kappa_1\alpha i}} \tau_{\kappa_1\alpha i} \right] \psi \quad (3)$$

Here ψ is the excess electron's wave function, $\tau_{\kappa\alpha i}$ the atomic displacement with $\kappa\alpha i$ denoting the Cartesian coordinate α of atom κ in the i th unit cell. H_{KS}^0 , $\frac{\partial V_{KS}^0}{\partial \tau_{\kappa_1\alpha i}}$, and $\psi_{n\mathbf{k}}^0$ are the ground state Kohn-Sham (KS) Hamiltonian, the variation of the KS potential, and the wave function with band index n and wave vector \mathbf{k} , respectively. $C_{\kappa_1\alpha i, \kappa_2\beta j}$ represents the force constant matrix. The total energy for an exciton-phonon coupled system can be expressed by replacing the KS Hamiltonian with the many-body BSE Hamiltonian and the electronic wave function by its exciton counterpart

$$E\{\tau_{\kappa\alpha i}, \psi_{ex}\} = E\{\tau_{\kappa\alpha i}^0, \psi_{ex}^0\} + \frac{1}{2} C_{\kappa_1\alpha i, \kappa_2\beta j} \tau_{\kappa_1\alpha i} \tau_{\kappa_2\beta j} + \int \psi_{ex}^* \left[H_{BSE} + \frac{\partial H_{BSE}}{\partial \tau_{\kappa_1\alpha i}} \tau_{\kappa_1\alpha i} \right] \psi_{ex}, \quad (4)$$

where ψ_{ex}^0 denotes the exciton empty state. The *BSE* Hamiltonian can be written as $H_{vc,v'c'}^{BSE} = (\varepsilon_c - \varepsilon_v)\delta_{cc'}\delta_{vv'} + (2V_{vc,v'c'} - W_{vc,v'c'})$, with W and V representing the direct electron-electron attraction and the exchange term, respectively. $c(c')$ and $v(v')$ are the conduction and valence band indices.

Applying the principles of energy minimization and exciton number conservation to Eq. 4, we reach the following self-consistent eigen equations [*Phys. Rev. B* 105, 085111 (2022); *Phys. Rev. Lett.* 125, 107401 (2020)]:

$$\frac{2}{N_p} B_{\mathbf{q}\mu} G_{nmv}^{ex-ph}(\mathbf{Q}, \mathbf{q}) A_{m\mathbf{Q}+\mathbf{q}} = (\varepsilon_{n\mathbf{Q}} - \varepsilon) A_{n\mathbf{Q}},$$

$$B_{\mathbf{q}\mu} = \frac{1}{N_p} A_{m\mathbf{Q}+\mathbf{q}}^* \frac{G_{nmv}^{ex-p}(\mathbf{Q}, \mathbf{q})}{\hbar\omega_{\mathbf{q}\mu}} A_{n\mathbf{Q}}. \quad (5)$$

Here N_p is the number of unit cells in the supercell. $A_{n\mathbf{Q}}$ denotes the wave function in the exciton basis, with $B_{\mathbf{q}\mu}$ the lattice wave function in the phonon eigenmode basis. The

exciton-phonon coupling matrix $G_{nmv}^{ex-p}(\mathbf{Q}, \mathbf{q})$ is defined by the differential of the *BSE* Hamiltonian with respect to lattice displacement along a phonon normal mode. Assuming constant static screening function, one can estimate the $G_{nmv}^{ex-ph}(\mathbf{Q}, \mathbf{q})$ using the exciton wave function in the electron-hole pair basis $E_{v\mathbf{k},c\mathbf{k}+\mathbf{Q}}^{n\mathbf{Q}}$ and electron-phonon coupling matrix g^{el-ph} by

$$G_{nm\mu}^{ex-ph}(\mathbf{Q}, \mathbf{q}) = E_{v\mathbf{k},c\mathbf{k}+\mathbf{Q}+\mathbf{q}}^{m\mathbf{Q}+\mathbf{q}*} E_{v\mathbf{k},c'\mathbf{k}+\mathbf{Q}}^{n\mathbf{Q}} g_{cc'\mu}^{el-ph}(\mathbf{k} + \mathbf{Q}, \mathbf{q}) - E_{v\mathbf{k}-\mathbf{q},c\mathbf{k}+\mathbf{Q}}^{m\mathbf{Q}+\mathbf{q}*} E_{v'\mathbf{k},c\mathbf{k}+\mathbf{Q}}^{n\mathbf{Q}} g_{vv'\mu}^{el-p}(\mathbf{k} - \mathbf{q}, \mathbf{q}), \quad (6)$$

where m and n denote the exciton band indices and \mathbf{Q} is the exciton center-of-mass momentum.

Following the Reviewer's kind suggestion, we have added a new section (Supplementary Note 13: Computational details) into the revised Supplementary Information (lines 190-236, pages 15 and 16), which includes the above discussion on how the coupling between the excitons and phonons is determined and how the exciton states are calculated.

In addition, we have added the following content in the revised manuscript (lines 180-184, page 5):

“By expanding the total energy of the coupled exciton-phonon system in a perturbation theory-based formula and applying the variational principles, we can self-consistently solve the eigenfunctions in the exciton and phonon basis, and obtain the mode- and momentum-resolved exciton-phonon coupling matrix element (see Methods and

Supplementary Note 13 for more details)".

(4) Dr. Prof. Emmanuel Rashba is a pioneer in the field of phonon-exciton coupling. Please add them to the citations. Along those lines, please have a concise paragraph that discusses exciton-phonon coupling in 2D systems. The fact that this is across the interlayer region is noteworthy. Exciton-phonon coupling in 2D systems is particularly interesting and your work builds onto this.

Response 5:

We fully agree with the Reviewer that Prof. Emmanuel Rashba is a pioneer in the field of phonon-exciton coupling. Following the Reviewer's kind suggestion, we have added a following paragraph to highlight the pioneering contributions of Prof. Rashba and to discuss exciton-phonon coupling in 2D systems in the revised manuscript (lines 212-218, page 6):

"We highlight that exciton-phonon coupling has been one of the research frontiers of condensed matter physics since its foundation in the 1950s^{58,59}. In 2D systems, exciton-phonon coupling has been extensively studied and is believed to underlie many intriguing physics, including but not limited to phonon-assisted dark-exciton formation^{60,61}, phonon-mediated valley depolarization^{62,63} and activation of optically silent phonon^{64,65}. However, previous work mainly focuses on the interaction between phonons and intralayer excitons. By contrast, our work reveals the exciton-phonon coupling involving highly tunable IXs, and uncovers the exotic phonon Stark effect".

Reference 58 [*Opt. Spectrosc.* 2, 75 (1957)] is the work of Prof. Rashba that proposed the exciton-phonon coupling;

Reference 59 [*Phys. Rev. B* 106, 210001 (2022)] is the work in honour of Prof. Rashba and his fundamental contributions to solid-state physics such as exciton-phonon coupling;

Reference 60: *Nat. Commun.* 11, 618 (2020);

Reference 61: *Nat. Commun.* 10, 2469 (2019);

Reference 62: *Phys. Rev. Lett.* 129, 027401 (2022);

Reference 63: *Nat. Commun.* 10, 807 (2019);

Reference 64: *Nat. Phys.* 13, 127-131 (2017);

Reference 65: *Phys. Rev. B* 99, 205410 (2019).

We thank the Reviewer for his/her very valuable reviewing efforts. We hope we have convincingly addressed all the comments raised by the Reviewer and our revised manuscript meets the criteria for publication in *Nature Communications*.

Reviewer #2 (Remarks to the Author)

The authors present a joint experiment-theory study on the appearance of Stark phonons in 2H-MoS₂ bilayers. These phonons appear as the IXs are tuned to resonate with the LA phonon emission line, and have a linear energy shift while tuning the applied out-of-plane electric field. Theoretical calculations are then carried out and find a strong coupling strength between IXs and LA phonons, which ‘dresses’ the phonons with exciton clouds and therefore fractions of electric dipole. Comparison with 3R-MoS₂ bilayer, where IXs and Stark phonons are missing, further proves their interpretation. Furthermore, a phonon intensity modulation as high as 1200% through an IX-mediated mechanism is observed as the IXs’ energies are tuned to the A_{2u} phonon line.

The addressed topic is novel and is in the core of one of the most active research areas in condensed matter physics. The manuscript is well written and the main message is clearly conveyed. The experimental data is beautiful and the theoretical calculations contribute to a better understanding of the exciton-phonon coupling in TMDC materials. So I think the work has the potential to deserve publication in Nature Communications. There are, however, a few issues that the authors need to address to convince me to recommend the work for publication:

Response 6:

We sincerely thank the Reviewer for his/her positive evaluation on our manuscript “*The addressed topic is novel..... The experimental data is beautiful..... the work has the potential to deserve publication in Nature Communications*”. We also appreciate the Reviewer’s insightful and constructive comments for improvement. Below we address the Reviewer’s comments point by point.

1) Why is MoS₂ A exciton (~1.92eV) missing from the spectra?

Response 7:

We thank the Reviewer for the comment. The obscurity of MoS₂ A exciton is because its intensity is much weaker than the intensities of both the phonons and interlayer excitons (IXs) under an on-resonance 633 nm laser excitation. Figure R1a shows the photoluminescence (PL) spectra of device H3 excited by 532 nm (black curve) and 633 nm lasers (red curve) under zero electric field. The A exciton of bilayer MoS₂ can be clearly distinguished around ~1.923 eV under an off-resonance 532 nm excitation. By contrast, the bilayer MoS₂ A exciton, although can be seen, is largely obscured by the strong signal of phonons under an on-resonance 633 nm excitation. Figure R1b shows the PL spectra at selected electric fields under 633nm excitation. Obviously, when the IX is tuned close to bilayer MoS₂ A exciton, its intensity is quite strong and would also largely obscure the A exciton.

To address this comment, we have added above discussions and Figure R1 into revised

Supplementary Information as Supplementary Note 12 (lines 179-189, page 14).

Fig. R1: **a**, PL spectra of device under 532 nm (black) and 633nm (red) excitation at zero electric field. **b**, PL spectra at selected electric fields under 633nm excitation.

2) If possible, it would be great to test with different laser wavelengths/different TMDC materials (Mo-based is preferred) to check if the Stark phonons are gone when the LA-IX coupling is absent.

Response 8:

We appreciate the Reviewer's the kind suggestion. In Fig. R2a, we present the Raman mapping of bilayer $2H$ -MoS₂ (device H1) as a function of electric field at 10 K with an off-resonance 532 nm laser excitation. Fig. R2b illustrates the first derivative of Fig. R2a. In contrast to the case of on-resonance 633 nm excitation in the main context, the LA-IX coupling is absent for the off-resonance 532 nm excitation, and no Stark phonon or enhancement of Raman signals is observed.

Following the Reviewer's kind suggestion, we fabricate new devices of hexagonal boron nitride encapsulated dual-gate bilayer $2H$ -MoSe₂ and bilayer $2H$ -WS₂. Figure R3a (R4a) shows the Raman mapping of a bilayer $2H$ -MoSe₂ ($2H$ -WS₂) device as a function of electric field under the 633 nm excitation. Figure R3b (R4b) illustrates the first derivative of Fig. R3a (R4a). Again, no noticeable signature of phonon Stark effect or intensity modulations is observed for both bilayer $2H$ -MoSe₂ and bilayer $2H$ -WS₂ where the LA-IX coupling is absent. This further confirms that the observed phonon Stark effect in bilayer $2H$ -MoS₂ results from the strong coupling between phonons and IXs.

To address this comment, we have added above discussions and Figs. R2-R4 into revised

Supplementary Information as Supplementary Note 11 (lines 152-178, pages 12 and 13).

Fig. R2: **a**, Raman spectra of bilayer $2H\text{-MoS}_2$ (device H1) under off-resonance 532 nm excitation as a function of electric field. **b**, First derivative of **a**.

Fig. R3: **a**, Raman spectra of a bilayer $2H\text{-MoSe}_2$ device under 633 nm excitation as a function of electric field. **b**, First derivative of **a**.

Fig. R4: **a**, Raman spectra of a bilayer $2H$ - WS_2 device under 633 nm excitation as a function of electric field. **b**, First derivative of **a**.

3) The theoretical calculation indicates a strong coupling between LA(M) phonon and IXs. Why is LA phonon's Raman intensity not modulated (to a high extent as A_{2u}) through the same IX-mediated mechanism?

Response 9:

We appreciate the Reviewer's comment. Given that the intensity of LA phonon is not strong, we have previously neglected its modulation with electric fields. Figure R5a shows the fitted emission intensities of LA phonon as a function of electric field. Clearly, LA phonon is activated when IXs are electrically tuned across its emission line. After activation, the intensity of LA phonon first increases and then decreases with the electric field. Since we cannot determine the LA phonon intensity under zero electric field, we define the electro-phonon modulation depth as $\rho = \frac{I - I_m}{I_m}$, where I (I_m) represents the phonon emission intensity at finite (maximum) electric field. Figure R5b presents the electro-phonon modulation depth ρ of LA phonon against the electric fields. The maximum modulation depth ρ of LA phonon mode can reach $\sim 800\%$. We remark that the modulation depth ρ of LA phonon is much larger than that of A_{1g} phonon, and only slightly smaller than that of A_{2u} phonon.

To address this comment, we have added above discussions and Fig. R5 into revised Supplementary Information as Supplementary Note 5 (lines 67-79, page 6).

Fig. R5: **a**, The fitted emission intensities of LA phonon as a function of electric field. **b**, The electro-phonon modulation depth ρ of LA phonon against electric fields.

4) Similar question to 3), why do A_{1g} and A_{2u} not have phonon Stark effect as their couplings to IXs are also strong? If possible, it would be great to elaborate in the manuscript what might determine the results (SP or intensity modulation) of the phonon-IX coupling.

Response 10:

We thank the Reviewer for the insightful comment. We are sorry that we do not yet well understand why A_{1g} and A_{2u} do not show phonon Stark effect, although we became aware of this since our initial measurements and have been trying to understand this. One possibility is that LA and A_{1g}/A_{2u} phonons are from different high symmetry points in the Brillouin zone, and show distinct symmetries and energies. Further in-depth investigation is required to solve this puzzle, which is beyond the scope of the present study.

5) In figure 1d & 2a, the peak at $1.923\text{eV}/295\text{cm}^{-1}$ at higher electric field seems to share similar features with the intensity-modulated A_{2u} phonon. Is this peak repeatable? (It seems absent from 3R-MoS₂. How about H2 and H3?)

Response 11:

We sincerely thank the Reviewer for pointing out this meticulous feature that we have overlooked. Following the Reviewer's comment, we have carefully checked the results of devices H2 and H3. However, the feature mentioned by the Reviewer in device H1 is absent in devices H2 (Fig. R6a) and H3 (Fig. R6b). Given that such meticulous feature isn't repeatable, it may not be intrinsic.

We thank the Reviewer for his/her very valuable reviewing efforts. We hope we have convincingly addressed all the comments raised by the Reviewer and our revised manuscript meets the criteria for publication in *Nature Communications*.

Fig. R6: First-order derivative of the Raman spectra of device H2 (a) and H3 (b). Green and navy dotted lines represent IXs and black dotted lines denotes SP modes.

Reviewer #3 (Remarks to the Author)

In this manuscript, Huang, et.al. explores an intriguing linear shift in energy and intensity modulation of phonon modes in bilayer 2H MoS₂ via an external electric field. The term Phonon Stark Effect is coined. Although experimentally detailed and computationally robust, the authors have failed to draw major insights based on their observation and certain technical aspects remain unclear. This makes it lacking the content and the quality required for publication in Nature Communication in its current form. Listed are few major technical queries regarding the results followed by some minor suggestions and general comments.

Response 12:

We sincerely thank the Reviewer's positive comment "experimentally *detailed and computationally robust*". We also appreciate the Reviewer's insightful and constructive comments for improvement. Below we address the Reviewer's comments point by point.

- What is the physical implication of a DC Phonon Stark effect in the bilayer MoS₂? In reference 38, vibrational Stark effect is introduced where the DC electric field modifies the intermolecular interactions via dipole-dipole interaction, which shows up as a vibrational energy renormalization. What is the counterpart in solid state system and why does it need an exciton to mediate it? All available theory (to my knowledge) on Stark effect for phonon dressed excitons refer to a AC Stark effect where exciton energy bands are renormalized due to an exchange type interaction [Example Phys. Rev. B 58, 1349,1998, phys. stat. sol. (b), 150: 431-436,1988]. But the effect described in the paper is renormalization of phonon energy state upon application of a DC field, which signifies some sort of material modification like stress or strain. Is that what is happening?

Response 13:

We fully agree with the Reviewer that for a molecule system, the interaction between the intermolecular dipole and an external DC electric field can produce a shift in the vibrational frequency and thus result in the vibrational Stark effect. However, in solid state systems, the direct coupling between phonon (which is a collective excitation) and DC electric field is generally ignorable because the net dipole is typically zero, especially for non-ferroelectric bilayer MoS₂ studied in our work. This suggests that the phonon Stark effect in bilayer MoS₂ should have a totally different origin, compared to the vibrational Stark effect previously observed in molecule systems. By combining experimental results and theoretical calculations, we demonstrate that the observed phonon Stark effect in bilayer 2H-MoS₂ should originate fundamentally from the strong coupling between LA(M) phonon with highly tunable interlayer excitons (IXs), which enables the formation of exciton-dressed phonon states with nonvanishing static dipole moment.

You are right. Previous studies focused on the phonon-dressed exciton states where exciton energies are renormalized by the exchange type interaction (i.e., phonon-mediated AC Stark effect). In our work, what we uncover is the exciton-dressed phonon states and we

demonstrate the linear shift of phonon energy upon application of a DC electric field.

Pertaining to the possibility of stress or strain mentioned by the Reviewer, we believe that this is not the case of our results. First, apart from the LA(M) phonon, other phonon modes, for example E_{2g} at $\sim 385 \text{ cm}^{-1}$ and A_{1g} at $\sim 405 \text{ cm}^{-1}$ that are sensitive to stress or strain, do not show energy shift with electric field. Second, no apparent phonon Stark effect or intensity modulation is observed under the same experimental conditions for bilayer 3R-MoS₂, which exhibits similar electronic structure and phonon dispersion to the bilayer 2H counterpart, but lacks the IXs. This largely rules out the possibility of stress or strain.

To further confirm that stress or strain does not underlie the observed phonon Stark effect, we performed the electric field-dependent Raman spectra of bilayer 2H-MoS₂ under 532 nm excitation (Fig. R7), bilayer 2H-MoSe₂ (Fig. R8) and bilayer 2H-WS₂ (Fig. R9) under 633 nm excitation. For these off-resonance excitation cases (i.e., phonon-IX coupling is absent), no noticeable signature of phonon Stark effect or intensity modulation is observed. This rules out the possibility of the strain mechanism, and confirms that the observed phonon Stark effect in bilayer 2H-MoS₂ results from the strong coupling between phonons and IXs.

To address this comment, we have added the following content in the revised manuscript (lines 209-211, page 6):

“It is noteworthy that previous studies mainly focused on the phonon-dressed exciton states where the energies of excitons, rather than phonons, are renormalized by the exchange type interaction, i.e., phonon-mediated AC Stark effect^{56,57”.}

Refs. 56 and 57 are the two papers mentioned by the Reviewer [Phys. Rev. B 58, 1349,1998, phys. stat. sol. (b), 150, 431-436,1988].

In addition, we have added the above discussions and Figs. R7-R9 into revised Supplementary Information as Supplementary Note 11 (lines 152-178, pages 12 and 13).

Fig. R7: **a**, Raman spectra of device H1 under 532 nm excitation as a function of electric field. **b**, First derivative of **a**.

Fig. R8: **a**, Raman spectra of a bilayer 2H-MoSe₂ device under 633 nm excitation as a function of electric field. **b**, First derivative of **a**.

Fig. R9: **a**, Raman spectra of a bilayer $2H$ - WS_2 device under 633 nm excitation as a function of electric field. **b**, First derivative of **a**.

- What roles does the resonance condition play? When IX is tuned over the Raman line, is there really a resonance condition met? Energetically phonons are far away from excitons. The exciton is tuned between 1.91 and 1.94 eV, while the 230cm^{-1} LA phonon line has an energy of 29meV?

Response 14:

We appreciate the Reviewer's comment. For a Stokes Raman scattering process (left panel of Fig. R10), an electron in ground state is first excited by an incident photon to an intermediate state E_1 with an energy equal to that of the excitation photon energy E_{ex} , and then scattered to another intermediate state E_2 by emitting a phonon with energy $h\omega_{ph}$ (i.e., the energy of intermediate state E_2 is $E_{ex} - h\omega_{ph}$). The electron in the intermediate state E_2 recombines with the hole in the ground state to emit a photon with energy of $E_{ex} - h\omega_{ph}$, which is detected by the Raman instrument. For Raman scattering process, the resonance condition is satisfied when the energy of exciton matches well with the intermediate state E_1 or E_2 .

In our case with a 633 nm excitation, the energy of intermediate state E_1 is ~ 1.959 eV, and the energy of intermediate state E_2 is ~ 1.93 eV given the LA phonon energy is ~ 0.029 eV. When the electric field $F_z = 0$, the IX has the energy of ~ 1.951 eV (green line, right panel of Fig. R10), which is 21 meV above intermediate state E_2 . As F_z increases, the energy of IX redshifts, and then a resonance Raman condition can be reached at $F_z = F_z^R$ when the energy of IX coincides with the Raman line (that is, the intermediate state E_2).

To address this comment, we have added the above discussions and Fig. R10 into revised Supplementary Information as Supplementary Note 7 (lines 87-105, page 8).

Fig. R10: Left panel: Schematic of a Stokes Raman scattering process. Right panel: Energy distributions of IX at three selected F_z .

- As a follow up question to the previous question, in the experiments how is the spectral filtering of excitonic PL and Raman signal done, especially when they become resonant? Because of the GW calculation backing, it is very evident that there is large exciton phonon coupling for certain phonon modes. But the fact that the energy renormalization requires the exciton line to be resonantly tuned to a Raman line begs the following question: Are we seeing some kind of optical Fano effect between the coherent Raman scattering signal and the PL from the exciton? That would explain the red shift of the 230 cm^{-1} Raman line compared to the blue shift of the IX_1 and IX_2 lines in cm^{-1} under applied DC bias F_z (Exciton red shifts and thus the energy delta between the Raman laser and the exciton peak increases and thus IX appear to blue shift in cm^{-1} scaling).

Response 15:

We sincerely thank the Reviewer for the insightful comment. In fact, the excitonic PL and Raman signal are recorded simultaneously and we do not do the spectral filtering. We fully agree with the Reviewer, the PL and Raman signal overlap when they become resonant, and therefore cannot be well distinguished. However, after the IX is tuned across the phonon line by electric field F_z via the quantum-confined Stark effect, the energy difference between them allows us to extract the PL and Raman signals by fitting.

Following the Reviewer's kind comment on the optical Fano effect, we have carefully checked the emission spectral lines of IXs (Fig. R11a) and LA phonon (Fig. R11b). Notably, both the emission spectral lines of IXs and LA phonon are symmetric, and can be perfectly fitted by Lorentz function. This is in stark contrast to the optical Fano effect which typically gives asymmetric Fano resonance curves. In addition, we highlight that the blue shift of the IX lines in cm^{-1} unit under applied electric field F_z is not due to the optical Fano effect,

but because IXs have an out-of-plane static electric dipole and thus show linear Stark shift under F_z [please refer to *Nat. Nanotechnol.* 15, 901-907 (2020) for more details]. Moreover, if the observed phenomena are due to the optical Fano effect, a change of the slope of IXs should be expected when IXs are tuned across the phonon line. This is not the case of our work where the slope of IXs has remained constant (please refer to Figs. 1d, 1e, 2a-2c in the main text for more details), ruling out the optical Fano effect.

To address this comment, we have added the above discussions and Fig. R11 into revised Supplementary Information as Supplementary Note 8 (lines 106-117, page 9).

Fig. R11: The emission spectral lines of IXs (a) and LA phonon (b) for device H2.

- An effective way to dispel my secondary explanation in the previous bullet point is to perform Raman with say 532nm excitation, where the 230cm⁻¹ LA phonon would be far from the IX tuning range. Do we expect this phonon Stark affect to show up? If not, then we need to clarify the type of resonance condition we are describing, as the phonon states are always present in the solid at finite temperatures. And if we generate excitons, because of physical proximity and presence of exchange type interactions, the excitons might be dressed with phonons. In other words, if spectral overlap of the IX PL and Raman is key to this phenomenon, then an explanation for how to exclude optical Fano effects need to be provided. Maybe this previous work will provide some insight. *ACS Nano*, 15, 9, 15371–15380, 2021. I am attaching an example of optical Fano resonance for plasmonic nanoparticles below.

Response 16:

Following the Reviewer's kind suggestion, we have performed the Raman mapping of bilayer $2H$ -MoS₂ (device H1) as a function of electric field at 10 K with a 532 nm laser excitation (please refer to Fig. R7). In contrast to the case of 633 nm excitation in the main context, no phonon Stark effect or enhancement of Raman signals is observed for the 532 nm excitation.

The lack of phonon Stark effect under 532 nm laser excitation can be understood as the lack of the LA-IX coupling. For the case with a 532 nm laser excitation, the energy of intermediate state E_1 is ~ 2.33 eV, and the energy of intermediate state E_2 is ~ 2.301 eV given the LA phonon energy is ~ 0.029 eV (please refer to Fig. R10). Because the energies of both intermediate states E_1 and E_2 are more than 350 meV above the IXs, the resonance Raman condition cannot be achieved by controlling the energies of IXs with electric field F_z , and therefore phonon Stark effect is absent.

We sincerely thank the Reviewer for referring the nice work of ACS Nano, 15, 15371 (2021) and attaching an example of optical Fano resonance for plasmonic nanoparticles. As our response 15 to your previous comment, we believe that the observed phonon Stark effect is not due to the optical Fano effect, but because of the formation of exciton-dressed phonon states induced by the strong coupling between LA(M) phonon and highly tunable IXs.

- Why do LA phonon and the A_{2u} phonon show different behavior upon DC field? LA phonon redshifts and the A_{2u} blueshifts.

Response 17:

We sincerely thank the Reviewer for the comment. We are sorry that we currently do not have a good understanding of why LA and A_{2u} phonons show different behaviours, although we became aware of this since our initial measurements and have been trying to understand this. One possibility is that LA and A_{2u} phonons are from different high symmetry points in the Brillouin zone, and show distinct symmetries and energies. Further in-depth investigation is required to solve this puzzle, which is however beyond the scope of present study.

Minor Suggestions

- Please include linewidths of the IX exciton PL in the text or supplementary information.

Response 18:

Following the Reviewer's kind suggestion, we have extracted the full width at half maximum (FWHM) of the IXs by fitting for devices H1 (black circles), H2 (red circles) and H3 (navy circles), as shown in Fig. R12. For all the three devices, the FWHM of the IXs is very narrow and only ~ 3.5 meV on average.

To address this comment, we have included the above discussions and Fig. R12 into revised Supplementary Information as Supplementary Note 6 (lines 80-86, page 7).

Fig. R12: Fitted FWHM of IXs in devices H1 (black circles), H2 (red circles) and H3 (navy circles).

In conclusion the work presented here currently does not meet the standards required for publication in Nature Communication in its current form. Some revisions along with more insightful discussions of this work is essential to be considered for publication.

Response 19:

We thank the Reviewer for his/her very valuable reviewing efforts. We hope we have convincingly addressed all the comments raised by the Reviewer and our revised manuscript meets the criteria for publication in *Nature Communications*.

REVIEWER COMMENTS

Reviewer #1 (Remarks to the Author):

Thank you for answering my questions and affecting changes. All the changes met my requests and I believe the manuscript is ready to be published.

Reviewer #2 (Remarks to the Author):

The authors have addressed all my comments in a convincing way. I especially thank the authors for confirming the missing SP using different excitations and TMDCs. And I also like the authors' response to the other reviewers. So I am happy to recommend publishing this work in Nature Communications.

Reviewer #3 (Remarks to the Author):

I thank the authors for the extremely detailed rebuttal for all the review comments. I appreciate the extra experiments done in response to the suggestions made by various reviewers. Unfortunately, I maintain my stance of skepticism of the spectral resonance induced movements of the phonon lines in bilayer MoS₂ and have made the decision to reject the manuscript. I will summarize my reasoning for my stance in two major points.

1.

a) In response to reviewer 1 a very detailed description of the GW-BSE calculations has been added to the Supplementary Information. In short, the authors use perturbative methods to account for renormalization of exciton-phonon coupling under addition of an electron (charge). Firstly, this calculation holds for explaining previous works of renormalization of phonon energies due to modified/perturbed electron-phonon coupling. The author has added multiple references of previous work that explore this method of phonon energy renormalization (Ref: 60-65). However, such perturbative technique doesn't hold for the dual gate (floating gate) geometry where no charge is assumed to be injected in the device and it functions in a capacitor geometry rather than a diode geometry.

b) If we were to overlook this point and assume that the analysis holds for floating gate geometry, the self-consistent equations have no energy resonant terms. The exchange interaction energy is dependent of k and Q , which are crystal momentum vectors. This makes sense as exciton-phonon coupling doesn't depend on the energetic position of the exciton resonance but rather the Bloch wavevectors, as expected from well established condensed matter theories for crystals. So, the requirement of spectral overlap of exciton and phonon energy for this phenomenon to occur is very non-intuitive and the authors have not provided a clear possible direction to look towards for further studies of this.

2.

a) The disappearance of this effect upon "non-resonant" excitation at 532nm shows another major roadblock towards my understanding of this phenomenon. Because if the phenomenon could have been reproduced at 532nm, this would make it consistent with the existing theoretical models which do not have any energy resonance terms in their Hamiltonians for GW-BSE type calculations.

b) On the note of the optical Fano effect, I will elaborate a little more as there are other nuances to consider. An excerpt from Wikipedia says the following: "*The Fano resonance line-shape is due to interference between two scattering amplitudes, one due to scattering within a continuum of states (the background process) and the second due to an excitation of a discrete state (the resonant process).*" Here the background process is the IX emission (broader resonance) and the dominant scattering process is the Raman scattering process. Fano resonance does not modify the continuum scattering profile and hence the IX PL slope should remain more or less linear as pointed out in the SI section 11. The line shape of the Raman scattering however depends on the

“q” factor of the Fano resonance expression. Since the Raman scattering process can be clearly discerned over the PL signature of the IX, the q factor is greater than one when IX is off resonance with the Raman line. As E-field is tuned to have the IX match resonance with the Raman line, q approaches $q < 1$. I am attaching some simulation snapshots showing the lines shapes as $q \rightarrow q < 1$ (q goes from 4 to 0.5, with $q = 4$ meaning Raman signal is 4x PL signal of IX). This shows two things, first when IX resonance is far away, and the intensity difference is high, the line shapes will be close to Lorentzian (No line cuts are shown in Supplementary figure 11a, 11b or 11c but we can refer to figure 2e in main manuscript where the asymmetric line shape can be seen in the line cuts). As the IX approaches resonance, an almost anti-crossing looking line shape will emerge as the PL intensity starts to match the Raman intensity. This manifests as a “giant peak shift” of the Raman line when IX resonance matches the Raman resonance.

The point of this comment is not to say that optical Fano resonance is the only answer to this observation. But the fact that this phenomenon can have an alternative explanation to this spectral overlap type modification of exciton-phonon exchange energy, makes me skeptical that this is truly an emergent phenomenon due to an interacting condensed matter system. **For this reason, I will have to reject the manuscript on the grounds of inconclusive inference of the results and the follow-up experiments failing to fully explain the phenomenon or disprove the alternative explanations.**

RESPONSE TO REVIEWERS' COMMENTS

Reviewer #1 (Remarks to the Author):

Thank you for answering my questions and affecting changes. All the changes met my requests and I believe the manuscript is ready to be published.

Response 1:

We sincerely thank the Reviewer for the positive assessment of our revisions and his/her recommendation for publication in *Nature Communications*.

Reviewer #2 (Remarks to the Author):

The authors have addressed all my comments in a convincing way. I especially thank the authors for confirming the missing SP using different excitations and TMDCs. And I also like the authors' response to the other reviewers. So I am happy to recommend publishing this work in Nature Communications.

Response 2:

We sincerely thank the Reviewer for the very positive evaluation of our revisions and his/her recommendation for publication in *Nature Communications*. We also appreciate the Reviewer for carefully checking our responses to other reviewers' comments and admiring them.

Reviewer #3 (Remarks to the Author):

I thank the authors for the extremely detailed rebuttal for all the review comments. I appreciate the extra experiments done in response to the suggestions made by various reviewers. Unfortunately, I maintain my stance of skepticism of the spectral resonance induced movements of the phonon lines in bilayer MoS₂ and have made the decision to reject the manuscript. I will summarize my reasoning for my stance in two major points.

Response 3:

We sincerely thank the Reviewer for the positive evaluation on our revisions and the extra experiments. We appreciate the Reviewer's new comments and address them below point by point.

1.

a) In response to reviewer 1 a very detailed description of the GW-BSE calculations has been added to the Supplementary Information. In short, the authors use perturbative methods to account for renormalization of exciton-phonon coupling under addition of an electron (charge). Firstly, this calculation holds for explaining previous works of renormalization of phonon energies due to modified/perturbed electron-phonon coupling. The author has added multiple references of previous work that explore this method of phonon energy renormalization (Ref: 60-65). However, such perturbative technique doesn't hold for the dual gate (floating gate) geometry where no charge is assumed to be injected in the device and it functions in a capacitor geometry rather than

a diode geometry.

Response 4:

We thank the Reviewer for the comments on our description of *GW-BSE* calculations, which has been added to Supplementary Information based on the kind requests of Reviewer #1 in the first round. The Reviewer had the impression that we calculate the renormalisation of exciton-phonon coupling from adding an electron. We apologize for the possible confusion arising from the description of our methods. In fact, as Reviewer #1 has mentioned, the fact that the exciton wavefunction is rather localized poses great challenges in calculating its coupling with phonons with a well-defined momentum q . Therefore, the perturbation framework allows us to use a self-consistent approach to manage the realistic and tractable computation of exciton-phonon coupling.

We would like to highlight that in our calculation, there is no need for carrier injection as the exciton wavefunction is computed through the state-of-art *GW-BSE* calculations. The calculated exciton-phonon coupling is an intrinsic property of bilayer MoS₂, which is relevant to their momentum vectors and energies while do not depend on the gate geometry. Meanwhile, we note that other work [*Phys. Rev. Lett.* 132, 036902 (2024); *Phys. Rev. B* 109, 045202 (2024); arXiv:2311.12662] has employed the same technique to calculate the exciton-phonon coupling.

We appreciate Reviewer's comments on the Refs. 60-65, which have been added to the revised manuscript to discuss the recent advance of exciton-phonon coupling in 2D systems based on the kind requests of Reviewer #1 in the first round. We would like to note that Refs. 60-65 added in the revised manuscript is to show that intralayer exciton-phonon coupling underlies many intriguing physics in 2D systems (such as phonon-assisted dark-exciton formation, phonon-mediated valley depolarization, and activation of optically silent phonon), rather than phonon energy renormalisation. In fact, Refs. 60-65 are not relevant to either phonon energy renormalisation or *GW-BSE* calculations.

For more clarity, we have changed the description of the *GW-BSE* calculations “we start from a perturbation theory-based framework, which has been applied to compute polaron formation^{8,9}, where the total energy of a system containing an extra electron can be described” into “we start from a perturbation theory-based framework^{17,18}, where the total energy of an electron-phonon coupled system can be described” in the revised Supplementary Information (line 276-278 in page 18).

b) If we were to overlook this point and assume that the analysis holds for floating gate geometry, the self-consistent equations have no energy resonant terms. The exchange interaction energy is dependent of k and Q , which are crystal momentum vectors. This makes sense as exciton-phonon coupling doesn't depend on the energetic position of the exciton resonance but rather the Bloch wavevectors, as expected from well established condensed matter theories for crystals. So, the requirement of spectral overlap of exciton and phonon energy for this phenomenon to occur is very non-intuitive and the authors have not provided a clear possible direction to look towards

for further studies of this.

Response 5:

We express our gratitude to the Reviewer for the insightful comments. Given the interconnected nature of this observation with the subsequent comment, we have chosen to address both jointly in the following response for a more cohesive and comprehensive clarification.

2. a) The disappearance of this effect upon “non-resonant” excitation at 532nm shows another major roadblock towards my understanding of this phenomenon. Because if the phenomenon could have been reproduced at 532nm, this would make it consistent with the existing theoretical models which do not have any energy resonance terms in their Hamiltonians for *GW-BSE* type calculations.

Response 6:

We thank the Reviewer for the above two comments. Since both the two are related to the resonance Raman scattering, we address them together.

We fully agree with the Reviewer that the calculated exciton-phonon coupling is an intrinsic property of bilayer MoS₂, which depends on the specific exciton state and phonon state, such as their momentum vectors and energies. However, we would like to note that we just calculate the strength of exciton-phonon coupling between exciton and phonon states, rather than the Raman scattering process. Consequently, our self-consistent equations do not contain the information of the excitation photon energies.

Experimentally, for a Raman scattering process, if we want to observe the effect of exciton-phonon coupling on the phonon states, the excitation photon energy should be taken into consideration. More specifically, the energy of the excitation photons must be close to the exciton energy, i.e., satisfaction of resonant Raman condition. Here we show more details:

Figure R1 schematically shows the Stokes Raman scattering process, an electron in the ground state is first excited by an incident photon to an intermediate state E_1 with an energy equal to that of the excitation photon energy E_{ex} , and then scattered to another intermediate state E_2 by emitting a phonon with energy $h\omega_{ph}$ (i.e., the energy of the intermediate state E_2 is $E_{ex} - h\omega_{ph}$). The electron in the intermediate state E_2 recombines with the hole in the ground state to emit a photon with energy of $E_{ex} - h\omega_{ph}$, which is detected by the Raman instrument (referred to as phonon emission line). Employing different excitation phonons (i.e., 633 nm excitation or 532 nm excitation), the corresponding energies of intermediate states E_1 and E_2 are completely different, as illustrated in the left panel of Fig. R1. Only when a suitable excitation laser is used and therefore the corresponding intermediate state E_1 or E_2 is close to the exciton state, can the exciton state involve in the Raman scattering process and thus affect the phonon state via exciton-phonon coupling.

In our case with a 633 nm excitation, the energy of the intermediate state E_1^{633} is

~ 1.959 eV, and therefore the energy of the intermediate state E_2^{633} (i.e., phonon emission line) is ~ 1.93 eV given that the LA phonon energy is ~ 0.029 eV. Considering that the interlayer exciton has the energy of ~ 1.951 eV at zero electric field, which is only ~ 0.021 eV above phonon emission line and can be easily tuned to resonate with it by electric fields, the interlayer exciton state can participate in the Raman scattering process and thus renormalise the energies or emission intensities of phonon states via strong exciton-phonon coupling. It is noteworthy that previous work has widely elucidated that the enhancement of phonon emission intensity or activation of Raman silent phonon due to exciton-phonon coupling can only occur under resonant excitation conditions [*Ann. Rev. Phys. Chem.* 27, 465 (1976); *ACS Nano* 8, 9629 (2014); *Phys. Rev. Lett.* 114, 136403 (2015); *Nano Lett.* 16, 2363 (2016); *Nat. Phys.* 13, 127 (2017); *Phys. Rev. B* 99, 205410 (2019)].

By contrast, for the case of a 532 nm laser excitation, the energy of the intermediate state E_1^{532} is ~ 2.33 eV, and the energy of the intermediate state E_2^{532} (i.e., phonon emission line) is ~ 2.301 eV given that the LA phonon energy is ~ 0.029 eV. Because the interlayer exciton state, as depicted in Fig. R1, is well below (more than 350 meV) the intermediate states E_1^{532} and E_2^{532} , it does not participate in the Raman scattering process, and thus we cannot observe the effect of exciton-phonon coupling on the phonon states.

To address this comment, we have added the above discussions and Fig. R1 into the revised Supplementary Information (lines 93-116, page 8).

Fig. R1: Left panel: Schematic of Stokes on- and off-resonant Raman scattering processes. Right panel: Energy distributions of IX at three selected F_z .

b) On the note of the optical Fano effect, I will elaborate a little more as there are other nuances to consider. An excerpt from Wikipedia says the following: “*The Fano resonance line-shape is due to interference between two scattering amplitudes, one due to scattering within a continuum of states (the background process) and the second due to an excitation of a discrete state (the resonant process).*” Here the background process is the IX emission (broader resonance) and the dominant scattering process is the Raman scattering process. Fano resonance does not modify the continuum scattering profile and hence the IX PL slope should remain more or less linear as pointed out in the SI section 11. The line shape of the Raman scattering however depends on the “q” factor of the Fano resonance expression. Since the Raman scattering process can be clearly discerned over the PL signature of the IX, the q factor is greater than one when IX is off resonant with the Raman line. As E-field is tuned to have the IX match resonance with the Raman line, q approaches q<1. I am attaching some simulation snapshots showing the lines shapes as q → q<1 (q goes from 4 to 0.5, with q= 4 meaning Raman signal is 4x PL signal of IX). This shows two things, first when IX resonance is far away, and the intensity difference is high, the line shapes will be close to Lorentzian (No line cuts are shown in Supplementary figure 11a, 11b or 11c but we can refer to figure 2e in main manuscript where the asymmetric line shape can be seen in the line cuts). As the IX approaches resonance, an almost anti-crossing looking line shape will emerge as the PL intensity starts to match the Raman intensity. This manifests as a “giant peak shift” of the Raman line when IX resonance matches the Raman resonance.

Response 7:

We thank the Reviewer for pointing out the optical Fano effect as a possible mechanism for the phonon Stark effect we observed. We fully agree with the Reviewer that optical Fano effect between the Raman scattering and a continuum background does have the potential to cause variations of the measured phonon energy. However, we believe that this is not our case. Please allow us to explain it in more details.

1) For Fano shape profile, the spectral shape can be described by the following equation [*Phys. Rev.* 124, 1866 (1961); *Rev. Mod. Phys.* 82, 2257 (2010); *Nat. Mater.* 11, 294 (2012); *2D Mater.* 4, 031007 (2017)]:

$$I(\omega) = I_0 \frac{[1 + (\omega - \omega_0)/(q\gamma)]^2}{1 + [(\omega - \omega_0)/\gamma]^2} \quad (1)$$

Here, I_0 (γ) is the intensity (FWHM) of uncoupled discrete state. $\frac{1}{q}$ denotes the Fano coupling strength, or more specifically the overlap, between the continuum and the discrete state. In the light of the Equation (1), we can derive the energy renormalisation

of the discrete state [*Rev. Mod. Phys.* 82, 2257 (2010); *Nat. Mater.* 11, 294 (2012)]:

$$\Delta\omega = \omega - \omega_0 = \frac{\gamma}{q} \quad (2)$$

As q approaches 0, a larger spectral overlap and thus a stronger Fano coupling strength is expected, giving rise to a larger energy renormalisation $\Delta\omega$.

Thus, if the observed phonon Stark effect originates from optical Fano effect, the maximum phonon energy renormalisation should take place when the emission line of interlayer exciton (IX) is tuned to just intersect the LA phonon line (i.e., the positions marked by solid cyan circles in Fig. R2, where the Fano coupling strength is strongest). By contrast, our results show that when the IX emission line is tuned to resonate with LA phonon line, the phonon mode just starts to redshift with the applied electric field, which indicates a negligible phonon energy renormalisation (Fig. R2). The apparent contradiction between our measured results and what is expected from optical Fano effect strongly rules out the optical Fano effect as a possible mechanism for the phonon Stark effect we observed.

Fig. R2 (adopted from Fig. 2 in the main text): Upper: Contour plot of the first-order derivative of Raman intensity as a function of phonon energy (bottom axis) and electric field F_z (left axis). Lower: Extracted phonon energy as a function of F_z . The solid cyan circles mark the positions where IX is tuned to intersect the LA phonon line.

2) In the light of the Equation (1), we can derive the FWHM of the discrete state [*Rev. Mod. Phys.* 82, 2257 (2010); *Nat. Mater.* 11, 294 (2012)]:

$$\text{FWHM} = \frac{\gamma(q^2+1)}{|q^2-1|} \quad (3)$$

Clearly, the FWHM of the discrete state is dependent on the q . Figure R3 shows the fitted FWHM of the Stark phonon (SP) state against the electric field F_z . The FWHM and thus q basically don't change with the electric field. Together with Equation (2), we can deduce that the phonon energy does not renormalise with the electric field in optical Fano scenario. This is in stark contrast to our observations and therefore rules out the

optical Fano effect as a possible mechanism for the phonon Stark effect we observed.

Fig. R3: The FWHM of the SP state as a function of electric field F_z .

3) As mentioned by the Reviewer, if the observed phonon Stark effect originates from optical Fano effect, an almost anti-crossing looking line shape should emerge when IX resonance matches the Raman resonance. However, we do not observe anti-crossing signature for all the devices we have measured. This also effectively excludes the possibility that the optical Fano effect underlies the phonon Stark effect we observed.

4) We highlight that the linear shift of phonon energy with the applied electric fields does not require the spectral overlap between IX and phonon states. Figure R4 shows the Raman spectra of device H2 at three selected electric fields, i.e., $F_z = -0.423$, -0.476 and -0.529 MV/cm. Clearly, although there is no noticeable spectral overlap between IX and phonon states, we can unequivocally distinguish the phonon Stark effect. This largely rules out the spectra overlap and therefore optical Fano effect as a possible mechanism for the phonon Stark effect we observed.

To address this comment, we have added above discussion into revised Supplementary Information as Supplementary Note 13 “Discussion on optical Fano effect” (lines 201-256, pages 15-17).

Fig. R4: On-resonant Raman spectra of device H2 at three selected F_z . Signals of Stark

phonon states (IXs) are marked by the grey dashed line (lime blue arrows). Note that each spectrum is acquired by subtracting the raw data at $F_z = 0$ to eliminate the influence of A exciton. There is no notable overlap between Stark phonon (SP) and IX, but SP still redshifts linearly with increasing F_z , ruling out spectra overlap as a possible mechanism of phonon Stark effect.

The point of this comment is not to say that optical Fano resonance is the only answer to this observation. But the fact that this phenomenon can have an alternative explanation to this spectral overlap type modification of exciton-phonon exchange energy, makes me skeptical that this is truly an emergent phenomenon due to an interacting condensed matter system. For this reason, I will have to reject the manuscript on the grounds of inconclusive inference of the results and the follow-up experiments failing to fully explain the phenomenon or disprove the alternative explanations.

Response 8:

We appreciate the Reviewer's insightful feedback and have rigorously excluded the possibility of Fano resonance effects in our findings. In addressing the concerns raised, we have expanded our analysis and experimental evidence to demonstrate the novelty of our observations, moving beyond simple spectral overlap explanations. Our updated manuscript now more convincingly supports the emergent phenomena within an interacting condensed matter system, aligning with standards of *Nature Communications*. We trust these revisions adequately address the Reviewer's points and make a strong case for publication.

REVIEWERS' COMMENTS

Reviewer #3 (Remarks to the Author):

I thank the authors once again for the detailed discussion and commend the extra efforts put in to drive the idea forward. I also appreciate the details put into disproving that optical Fano effect is not at play in this phenomenon. The requirement for spectral resonance of the Raman scattering energy and the IX energy still holds my skepticism. Especially when already pointed out in the rebuttal as *"We fully agree with the Reviewer that the calculated exciton-phonon coupling is an intrinsic property of bilayer MoS₂, which depends on the specific exciton state and phonon state, such as their momentum vectors and energies. However, we would like to note that we just calculate the strength of exciton-phonon coupling between exciton and phonon states, rather than the Raman scattering process. Consequently, our self-consistent equations do not contain the information of the excitation photon energies."*

The criteria provided for the requirement of spectral resonance is not backed by the BSE+GW calculations and I understand that it is indeed a very hard quantity to calculate and is likely beyond the scope of the current work. But in the light of the diligence that is put into the explanations, and the detail in the experiment and theoretical aspects of the work (also the difficulty of making the devices with existing 2D material transfer techniques), **I accept the manuscript for publication.** While I maintain my skepticism on the origins of the phenomenon, the work presented here deserves publication based on its novelty and the potential to break ground for further study of this phenomenon, in its current framework or another.

RESPONSE TO REVIEWERS' COMMENTS

Reviewer #3 (Remarks to the Author):

I thank the authors once again for the detailed discussion and commend the extra efforts put in to drive the idea forward. I also appreciate the details put into disproving that optical Fano effect is not at play in this phenomenon. The requirement for spectral resonance of the Raman scattering energy and the IX energy still holds my skepticism. Especially when already pointed out in the rebuttal as “We fully agree with the Reviewer that the calculated exciton-phonon coupling is an intrinsic property of bilayer MoS₂, which depends on the specific exciton state and phonon state, such as their momentum vectors and energies. However, we would like to note that we just calculate the strength of exciton-phonon coupling between exciton and phonon states, rather than the Raman scattering process. Consequently, our self-consistent equations do not contain the information of the excitation photon energies.”

The criteria provided for the requirement of spectral resonance is not backed by the BSE+GW calculations and I understand that it is indeed a very hard quantity to calculate and is likely beyond the scope of the current work. But in the light of the diligence that is put into the explanations, and the detail in the experiment and theoretical aspects of the work (also the difficulty of making the devices with existing 2D material transfer techniques), **I accept the manuscript for publication.** While I maintain my skepticism on the origins of the phenomenon, the work presented here deserves publication based on its novelty and the potential to break ground for further study of this phenomenon, in its current framework or another.

Response:

We sincerely thank the Reviewer for the positive evaluation on our revisions and his/her recommendation for publication in *Nature Communications*.